# The efficacy of *Lacticaseibacillus paracasei* MSMC39-1 and Bifidobacterium animalis TA-1 probiotics in modulating gut microbiota and reducing the risk of the characteristics of metabolic syndrome: A randomized, double-blinded, placebo-controlled study

Wongsakorn Luangphiphat[1,2], Pinidphon Prombutara[3,4], Praewpannarai Jamjuree[5], Chantanapa Chantarangkul[5], Porntipha Vitheejongjaroen[5], Chantaluck Muennarong[5], Krittapat Fukfon[6], Manasvin Onwan[7,8], Malai Taweechotipatr[5,8,9]*

1 Princess Srisavangavadhana College of Medicine, Chulabhorn Royal Academy, Bangkok, Thailand, 2 Division of Cardiology, Department of Medicine, Chulabhorn Hospital, Chulabhorn Royal Academy, Bangkok, Thailand, 3 Omics Sciences and Bioinformatics Center, Faculty of Science, Chulalongkorn University, Bangkok, Thailand, 4 Mod Gut Co., Ltd, Bangkok, Thailand, 5 Center of Excellence in Probiotics, Srinakharinwirot University, Bangkok, Thailand, 6 Boromarajonani College of Nursing Phayao, Faculty of Nursing, Praboromarajchanok Institute, Phayao, Thailand, 7 Department of Preventive and Social Medicine, Faculty of Medicine, Srinakharinwirot University, Ongkharak, Nakhon Nayok, Thailand, 8 Clinical Research Center, Faculty of Medicine, Srinakharinwirot University, Ongkharak, Nakhon Nayok, Thailand, 9 Department of Microbiology, Faculty of Medicine, Srinakharinwirot University, Bangkok, Thailand

* malai@g.swu.ac.th

## Abstract

Modern treatment, a healthy diet, and physical activity routines lower the risk factors for metabolic syndrome; however, this condition is associated with all-cause and cardiovascular mortality worldwide. This investigation involved a randomized controlled trial, double-blind, parallel study. Fifty-eight participants with risk factors of metabolic syndrome according to the inclusion criteria were randomized into two groups and given probiotics (*Lacticaseibacillus paracasei* MSMC39-1 and *Bifidobacterium animalis* TA-1) (n = 31) or a placebo (n = 27). The participants had a mean age of 42.29 ± 7.39 and 43.89 ± 7.54 years in the probiotics and placebo groups, respectively. Stool samples, anthropometric data, and blood chemistries were taken at baseline and at 12 weeks. The primary outcome was achieved by the probiotics group as their low-density lipoprotein-cholesterol level dramatically lowered compared to the placebo group (the difference was 39.97 ± 26.83 mg/dl, p-value <0.001). Moreover, significant reductions in body weight, body mass index, waist circumference, systolic blood pressure, and total cholesterol were observed in the volunteers treated with probiotics compared to the placebo. In the gut microbiome analysis, the results showed statistically significant differences in the beta diversity in the post-intervention probiotics group. *Blautia*, *Roseburia*, *Collinsella*, and *Ruminococcus* were among the gut microbiomes that were more prevalent in the post-intervention probiotics group. In addition, this group exhibited increases in the predicted functional changes in ATP-binding cassette (ABC) transporters,

**Data Availability Statement:** The raw sequence data are available under BioSample SAMN42867309 and BioProject PRJNA1141093.

**Funding:** the study was supported by the Chiangmai Bioveggie and National Innovation Agency (grant number PE0201-02-64-11-0252) and the Center of Excellence in Probiotics at Srinakharinwirot University (grant number 324/2565). The funders had no role in study design, data collection and analysis, decision to publish, or preparation of the manuscript.

**Competing interests:** I have read the journal's policy and the authors of this manuscript have the following competing interests: Pinidphon Prombutara is employed by Mod Gut Co., Ltd. This does not alter our adherence to PLOS ONE policies on sharing data and materials. The remaining authors declare that the research was conducted in the absence of any commercial or financial relationships that could be construed as a potential conflict of interest.

as well as ribonucleic acid transport, the biosynthesis of unsaturated fatty acids, glycerophospholipid metabolism, and pyruvate metabolism. In conclusion, this research demonstrated that the probiotics *L. paracasei* MSMC39-1 and *B. animalis* TA-1 have the efficacy to lower risk factors associated with metabolic syndrome.

## Introduction

Metabolic syndrome is characterized by a cluster of clinical features and metabolic abnormalities, including dyslipidemia, insulin resistance, hypertension, and abdominal obesity, which together increase the risk of developing cardiovascular disease and type 2 diabetes mellitus [1, 2]. It affects approximately 20% of the population, impacting millions of lives globally [3]. Many countries encourage their citizens to modify their dietary habits and engage in regular exercise to prevent and reduce the risk factors associated with metabolic syndrome [4]. Despite these efforts, the condition remains linked to increased all-cause and cardiovascular mortality [5].

The gastrointestinal tract is colonized by the intestinal microbiota, a key environmental factor that directly influences host health and contributes to the exacerbation of various diseases [6–9]. Dysbiosis, an imbalance in the microbiota, reduces its diversity and function, leading to metabolic issues [10]. A high-fat diet can worsen dysbiosis by increasing circulating lipopolysaccharide (LPS) and lipid levels, as well as compromising the intestinal barrier [11]. Additionally, dysbiosis may contribute to chronic inflammation and the development of obesity, diabetes mellitus, and metabolic syndrome through interactions between genetic and environmental factors [12, 13]. Treatment for metabolic syndrome and obesity-related metabolic endotoxemia involves preventing microbial dysbiosis and maintaining intestinal barrier integrity [14].

According to Hill et al. [15], the consensus statement that the International Scientific Association for Probiotics and Prebiotics (ISAPP) proposed for the proper usage of the term probiotic refers to live microorganisms that, when administered in adequate amounts, confer a health benefit on the host. Commercial strains are more likely to contain probiotics from the genera *Lactobacillus* and *Bifidobacterium* [16]. The key health benefits of probiotics include immune modulation, enhanced mineral and vitamin absorption, constipation relief, microbiome regulation, post-antibiotic microbiome stabilization, increased gastrointestinal resistance to pathogens, and pathogen reduction through short-chain fatty acid (SCFA) production [17–19].

Probiotics offer a potential strategy for managing body weight as well as metabolic syndrome by modulating the gut microbiome [20, 21]. They help maintain microbial balance and may reduce the risk of metabolic syndrome by controlling inflammation. A meta-analysis showed that probiotics, compared to placebos, significantly reduce body weight, fat mass, and the body mass index (BMI), supporting their role in improving metabolic health [22, 23]. Probiotic species like *Bifidobacterium longum*, *Lactobacillus acidophilus*, and *Lactobacillus gasseri* have demonstrated benefits by enhancing lipid metabolism, upregulating carbohydrate transport and metabolism genes, and improving digestion. They also modulate bile acid metabolism, reduce plasma glucose, cholesterol, and triglycerides, promote bile salt deconjugation, and aid in weight loss [24]. A 12-week mice study confirmed that *L. paracasei* HII01, xylo-oligosaccharides (XOS), and synbiotics improved dyslipidemia and insulin sensitivity, enhancing metabolic functions [25]. Similarly, a randomized placebo-controlled trial found *L. plantarum*

PBS067, *L. acidophilus* PBS066, and *L. reuteri* PBS072, along with prebiotics significantly reduced glucose, lipids, and inflammatory mediators in metabolic syndrome patients [26].

Interestingly, few clinical studies demonstrate the effect of probiotics on metabolic syndrome. Therefore, a double-blinded, randomized controlled trial was conducted to evaluate the efficacy of probiotics *L. paracasei* MSMC39-1 and *B. animalis* TA-1 in modulating gut microbiota and reducing the risk factors of metabolic syndrome.

## Materials and methods

### Subject enrollment

This study was a randomized, double-blind, controlled, parallel-design trial. The study population comprised 60 participants who met the inclusion criteria for metabolic syndrome risk factors and were recruited from the outpatient clinic at the Faculty of Medicine, Srinakharinwirot University, between September 29, 2022, and April 5, 2023. Stratified permuted block randomization was utilized to assign the participants into two groups. Group 1 received probiotics, consisting of *Lacticaseibacillus paracasei* MSMC39-1 (formerly *Lactobacillus paracasei*) at a dose of $1.0 \times 10^9$ colony-forming units (CFU) and *Bifidobacterium animalis* TA-1 ($1.0 \times 10^9$ CFU), along with a vegetable-based pellet. Group 2 received a placebo vegetable-based pellet. The participants in each group consumed five tablets (250 milligrams [mg]) per tablet), totaling 1,250 mg daily, 60 minutes before breakfast, for 12 weeks. The tablets were manufactured at a good manufacturing practice (GMP) certified food production facility. Probiotics with a vegetable-based pellet and a placebo vegetable-based pellet were compressed into tablets using microcrystalline cellulose (MCC) as a binder to enhance cohesion. The tablets were coated with hydroxypropyl methylcellulose (HPMC) to protect against moisture.

After being randomized into two groups, the participants' history, physical examination, blood test, and fresh stool were recorded at baseline and 12 weeks after the intervention. In this double-blind study, the probiotic and placebo capsules were identical in terms of appearance and packaging, guaranteeing that neither the volunteers nor the researchers knew who was receiving which medication.

During the study, the researcher called the participants to follow up each week for two weeks. Then, they followed up every two weeks until 12 weeks to check that the participants were demonstrating good compliance in taking the probiotics or the placebo, as this must be at a level of 80% or more.

The investigation was conducted following the Declaration of Helsinki and was approved by the ethics committee of Srinakharinwirot University (SWUEC/F-212/2565, September 29, 2022). The registration number for this study is TCTR20230505002. The probiotic strains used in this test can reduce sugar and lipid levels in both in vitro and in animal models [27–29]. The probiotic strains were approved for use as probiotic microorganisms in food by the Thai Food and Drug Administration (FDA), according to the Ministry of Public Health announcement. Written informed consent was obtained from all subjects involved in the research.

To assess participants' tolerance and satisfaction with the intervention, a quality-of-life questionnaire was administered at the final visit (12 weeks). The seven items included: (1) frequency of defecation (5 points for daily, down to 1 point for none); (2) difficulty in defecation (5 points for never, down to 1 for always); (3) stomach pain (5 points for never, down to 1 for always); (4) bloating (5 points for never, down to 1 for always); (5) use of laxatives (5 for never, down to 1 for always); (6) duration of defecation (5 points for <5 minutes, down to 1 for >30 minutes); (7) overall satisfaction (5 points for strongly agree, down to 1 for strongly disagree). This evaluation provided insights into participants' gastrointestinal health and satisfaction with the intervention.

## Inclusion criteria

The inclusion criteria for the participants in this study required individuals to meet the following conditions: (1) be within an age range of 18 to 60 years, and (2) exhibit three or more of the following criteria: (a) an elevated waist circumference ($\geq$102 cm in men and $\geq$88 cm in women), (b) elevated blood pressure ($\geq$130/85 mmHg), (c) reduced high-density lipoprotein cholesterol (HDL-C) levels (<40 mg/dL in men and <50 mg/dL in women), (d) elevated triglycerides ($\geq$150 mg/dL), or (e) elevated fasting blood glucose (FBG) (100–125 mg/dL) [1].

## Exclusion criteria

Participants who met the following criteria were excluded: (1) they had been diagnosed with the following underlying diseases: diabetes mellitus (hemoglobin A1c [HbA1c] $\geq$ 6.5 mg%) [30], hypertension (systolic blood pressure [SBP] > 140 mmHg, diastolic blood pressure [DBP] > 90 mmHg) [31], chronic kidney disease stage 3 and above or an estimated glomerular filtration rate (eGFR) value < 60 ml/min/1.73$^2$ [32], liver disease (aspartate transaminase [AST]/ alanine transaminase [ALT] > 5 times the normal value or jaundice), cancer, coronary artery disease (CAD), thyroid disease, immunodeficiency, or intestinal diseases such as inflammatory bowel disease, (2) they had used of antibiotics, immunosuppressants, probiotics supplements, synbiotics, herbal supplements, antacids or laxatives within 12 weeks before participating in the study, (3) smoked; (4) had alcoholism, (5) were pregnant or lactating, or (6) had acquired a Covid-19 infection within four weeks of the trial.

## Withdrawal or termination criteria

Participants were withdrawn from the investigation if they met any of the following criteria: (1) they received antibiotics, laxatives, or immunosuppressive drugs during the study period, (2) they experienced side effects from probiotics, including nausea, vomiting, diarrhea, facial swelling, or shortness of breath, as well as rashes or swollen lips, (3) were diagnosed with a non-communicable disease during the evaluation period, or (4) chose to withdraw.

## Sample size calculation

Low-density lipoprotein cholesterol (LDL-C) is considered a critical contributor to the development of cardiovascular disease [33]. For the present study, the primary outcome, changes in LDL-C, was used to determine the sample size, informed by previous research comparing placebo and intervention groups [34]. The placebo group exhibited a change of 0.17 ± 0.46 mmol/L, while the intervention group showed a difference of -0.40 ± 0.70 mmol/L. Using STATA software, the sample size was calculated with α = 0.05 and a power (1-β) = 0.90, resulting in 24 participants per group. To account for potential missing data (approximately 20%), the sample size was increased to approximately 28 participants per group.

## Sample collection and high-throughput sequencing

Blood samples were taken from each participant to measure the following parameters: FBG, HbA1c, total cholesterol, triglyceride, LDL-C, HDL-C, creatinine, AST, and ALT at baseline and 12 weeks post-intervention in both groups. At baseline and 12 weeks following the intervention, fresh stool was collected in deoxyribonucleic acid (DNA)/ribonucleic acid (RNA) shield fecal collection tubes (Zymo Research, USA). It was then promptly frozen at -20˚C for 48 hours before additional analysis. The QIAamp DNA Stool Mini Kit (Qiagen, USA) was used to extract DNA [35]. The quantity and quality of the DNA were assessed using nanodrop and electrophoresis [36]. By using 2X KAPA hot-start ready mix and 515 F and 806R primers,

the V4 hypervariable region of the 16S rRNA gene was amplified by polymerase chain reaction (PCR) [37].

The initial denaturation at 94˚C for three minutes was followed by 25 cycles of 98˚C for 20 seconds, 55˚C for 30 seconds, 72˚C for 30 seconds, and a final extension step at 72˚C for five minutes in the PCR. After eight cycles of the previously described PCR condition, the 16S amplicons were purified using AMPure XP beads and indexed with a Nextera XT Index Kit [38]. Ultimately, the PCR products underwent cleaning and pooling in preparation for the Illumina® MiSeq™250-base paired-end read sequencing and cluster formation [39, 40].

## Sequencing data analysis

The microbiome bioinformatics was analyzed using QIIME 2 2019.10. The q2-demux plugin was utilized to demultiplex the raw sequence data, and DADA2 (via q2-dada2) was employed to exclude reads that had expected errors (maxEE) greater than 3.0. With the SEPP q2-plugin, a phylogeny was built by inserting brief sequences into the reference phylogenetic tree, sepp-refs-gg-13-8.qza. Following the rarefaction of samples to a minimum read, the alpha-diversity metric, beta-diversity meter, and principle coordinate analysis (PCoA) were computed with q2-diversity [41]. The classify-sklearn naive Bayes taxonomy classifier was adopted to assign taxonomy to amplicon sequence variants (ASVs) based on comparison with the Greengenes 13_8 99% operational taxonomic units (OTUs) reference sequences [42]. By applying the Kruskal-Wallis test and a permutational multivariate analysis of variance (PERMANOVA) with 999 permutations, respectively, statistical tests of the alpha and beta diversity were carried out [43]. A heat tree analysis was generated for pairwise comparisons of the taxonomic differences between microbial communities with the MicrobiomeAnalyst web-based platform [44].

## Functional pathway enrichment analysis

The metabolic functions and pathways of the gut microbiome in each group were predicted with the PICRUSt2 software to explore their role in the metabolism [45]. Next, we utilized the STAMP software [46] to detect the differences in the metabolic function abundance between the groups. This was completed using Welch's t-test, a statistical method that compares the means of two groups to determine if there is a significant difference, with a confidence interval of 0.95 and a significance threshold of a corrected p-value <0.05.

## Statistical analysis and visualization

Stata/SE 16.1 software (StataCorp LP, College Station, TX, USA) was employed to analyze the statistical data [47]. Statistics were considered significant when the p-value < 0.05. All the study variables were subjected to descriptive statistics analysis, which was provided as the frequency (%) for categorical data and mean ± standard deviation (SD) or median for nonnormal quantitative data. The independent t-test with a p-value < 0.05 was utilized if the distribution of the quantitative data, such as age and laboratory results, was normal. The Mann-Whitney U test with a p-value <0.05 was applied if the data distribution was not normal. The Chi-square test or Fisher's exact test was used to compare the two groups for categorical data. The Benjamini-Hochberg (FDR) correction was applied to p-values for multiple hypothesis testing [48].

## Results

Sixty participants were randomized into either the probiotics (*Lacticaseibacillus paracasei* MSMC39-1 and *Bifidobacterium animalis* TA-1) or the placebo group. Notably, during the study, two participants from the placebo group were lost to follow-up, resulting in the analysis

CONSORT 2010 Flow Diagram

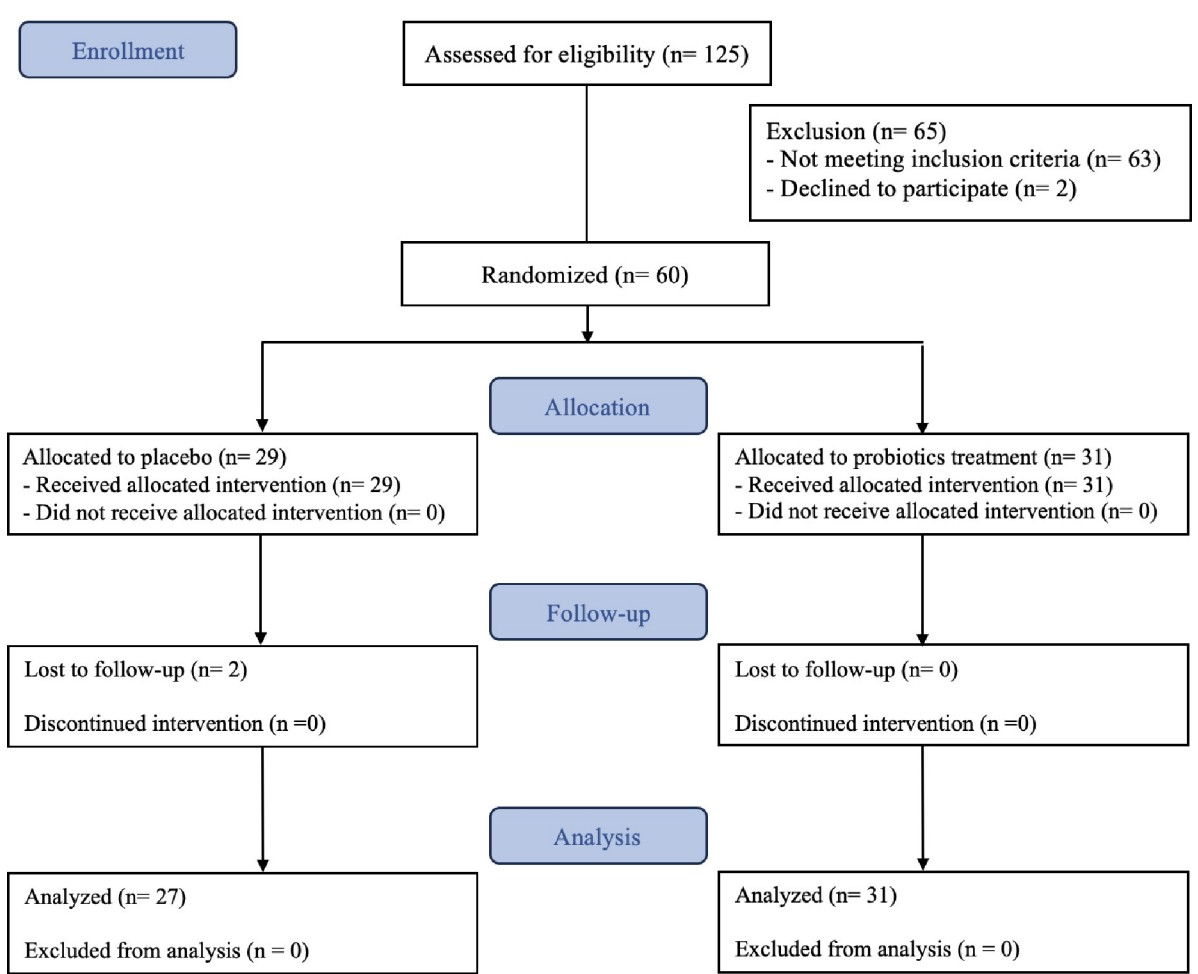

**Fig 1. The study flow diagram.**

of 58 blood and gut microbiota samples. No participants experienced any adverse events (Fig 1). Both groups consisted of approximately 70% female participants, with a mean age of 42.29 ± 7.39 years in the probiotics group and 43.89 ± 7.54 years in the placebo group. There was no statistically significant difference in age or gender among the groups (S1 Table).

Following the administration of the probiotic strains *L. paracasei* MSMC39-1 and *B. animalis* TA-1 over 12 weeks, a significant reduction was observed in the participants' body weight, BMI, waist circumference, SBP, and DBP when compared to the placebo group at baseline and 12 weeks. Additionally, the total cholesterol, triglycerides, LDL-C, and HbA1c levels showed statistically significant decreases from the baseline in the probiotics group by the end of the 12 weeks. However, no significant changes were observed in the placebo group (Table 1).

The differences between the pre-intervention and post-intervention results in the placebo and probiotics groups at 12 weeks are presented in Table 2. Following the administration of both strains of probiotics, namely *L. paracasei* MSMC39-1 and *B. animalis* TA-1 at 12 weeks, the difference in participants' body weight, BMI, and waist circumference, all significantly

**Table 1. Clinical and laboratory characteristics of the probiotics and placebo groups at baseline and week 12.**

| Variables | Placebo (n = 27) | | P-value | Probiotics (n = 31) | | P-value |
|---|---|---|---|---|---|---|
| | Baseline | Week 12 | | Baseline | Week 12 | |
| Weight (kg) | 75.90 (63.5–79.1) | 73.5 (61.3–79.2) | 0.089[2] | 76.47 ± 21.72 | 72.38 ± 21.23 | <0.001[1] |
| BMI (kg/m$^2$) | 27.07 (24.06–30.60) | 27.02 (24.39–30.18) | 0.070[2] | 28.27 ± 6.65 | 26.74 ± 6.56 | <0.001[1] |
| Waist circumference (cm) | 34.5 (32.5–38) | 34 (32–37.5) | 0.024[2] | 34.5 (30–39) | 32 (28–36.5) | <0.001[1] |
| Hip circumference (cm) | 40.5 (39.5–43.5) | 40.5 (38.5–43.5) | <0.001[2] | 41 (39–44) | 39 (37–42) | <0.001[1] |
| SBP (mmHg) | 120 (110–127) | 119 (110–127) | 0.826[2] | 127 (115–137) | 115 (108–124) | <0.001[2] |
| DBP (mmHg) | 80 (73–86) | 74 (70–80) | 0.002[2] | 82 (72–87) | 77 (69–82) | 0.007[2] |
| Total cholesterol (mg/dl) | 239.11 ± 29.85 | 235.11 ± 30.49 | 0.463[1] | 242 (218–262) | 194 (186–219) | <0.001[2] |
| Triglyceride (mg/dl) | 113 (84–168) | 107 (80–161) | 0.421[2] | 127.84 ± 55.57 | 109.32 ± 41.85 | 0.001[1] |
| HDL-C (mg/dl) | 54 (47–74) | 55 (47–69) | 0.791[2] | 56.65 ± 14.37 | 61.19 ± 16.05 | 0.001[1] |
| LDL-C (mg/dl) | 168.63 ± 29.68 | 163.67 ± 27.31 | 0.252[1] | 171 (146–188) | 124 (118–134) | <0.001[2] |
| FBG (mg/dl) | 92.67 ± 11.35 | 89.48 ± 9.20 | 0.111[1] | 92 (87–99) | 89 (83–98) | 0.543[2] |
| HbA1c (mg%) | 5.59 ± 0.39 | 5.54 ± 0.35 | 0.260[1] | 5.63 ± 0.41 | 5.49 ± 0.40 | 0.002[1] |
| Creatinine (mg/dl) | 0.71 (0.62–0.89) | 0.79 (0.67–1.04) | 0.008[2] | 0.75 ± 0.14 | 0.81 ± 0.14 | 0.001[1] |
| eGRF (ml/min/1.73$^2$) | 107.73 (89.53–111.57) | 96.78 (79.05–105.53) | 0.009[2] | 103.42 ±12.35 | 97.02 ± 14.23 | 0.004[1] |
| AST (IU/L) | 19 (16–22) | 21 (17–24) | 0.129[2] | 19 (16–26) | 18 (17–26) | 0.998[2] |
| ALT (IU/L) | 16 (12–24) | 19 (11–28) | 0.524[2] | 18 (13–27) | 16 (14–25) | 0.399[2] |

[1] Paired t-test

[2] Wilcoxon signed rank test; ALT, alanine aminotransferase; AST, aspartate aminotransferase; BMI, body mass index; DBP, diastolic blood pressure; FBG, fasting blood glucose; HbA1c, hemoglobin A1c; HDL-C, high-density lipoprotein cholesterol; IU/L, international units per liter; kg, kilogram; LDL-C, low-density lipoprotein cholesterol; mmHg, millimeters of mercury; mg, milligrams; mg/dl, milligrams per deciliter; SBP, systolic blood pressure

decreased from the baseline (3.90 kg, 1.36 kg/m$^2$, and 2.00 cm, p-value <0.001 for body weight, BMI, and waist circumference, respectively), and the variation differed significantly from the placebo group (Table 2). Moreover, following probiotics administration, the participant's SBP levels were significantly lower.

The total cholesterol, triglycerides, and LDL-C levels, considerably decreased after taking probiotics compared to before with statistical significance (the differences were 38.84 ± 27.47 mg/dl, p-value <0.001; 9 mg/dl, p-value 0.011; and 39.97 ± 26.83 mg/dl, p-value <0.001, respectively). Specifically, the primary outcome of this study, the LDL-C level, experienced a significantly high reduction in the probiotics group with statistical significance at 12 weeks (Table 2).

The FBG level considerably dropped after taking probiotics compared to previously in the same group, but it did not change significantly from the placebo group. The majority of participants, however, had normal FBG and HbA1c before the experiment. As a result, there was no discernible difference between the probiotic-treated and control groups (Table 2).

Following the intervention in the probiotics group, reductions in both male and female participants were also observed in total cholesterol, triglycerides, LDL-C, and HbA1c levels, along with an increase in HDL-C, when compared to the placebo group (S2 and S3 Tables).

Multivariate regression analysis revealed that the clinical and laboratory characteristics, as well as changes in variables between the probiotics and placebo groups, remained consistent. After 12 weeks of administration of the probiotic strains *L. paracasei* MSMC39-1 and *B. animalis* TA-1, a significant reduction in LDL-C after taking probiotics (39.97 ± 26.83 mg/dl) was achieved (p-value <0.001). Additionally, compared to the placebo group, participants in the probiotics group showed significant reductions at 12 weeks from baseline in body weight

**Table 2. The changes in 12 weeks of the intervention and control groups.**

| Variables | Placebo (n = 27) | Probiotics (n = 31) | P-value |
|---|---|---|---|
| Weight (kg) | -0.90 (-2.00, 1.10) | -3.90 (-4.70, 3.00) | <0.001[2] |
| BMI (kg/m$^2$) | -0.37 (-0.65, 0.36) | -1.36 (-1.75, 1.07) | <0.001[2] |
| Waist circumference (cm) | -0.50 (-2.00, 0.00) | -2.00 (-3.00, -2.00) | <0.001[2] |
| Hip circumference (cm) | -0.60 ± 0.62 | -2.21 ± 0.94 | <0.001[1] |
| SBP (mmHg) | -1 (-8, 7) | -10 (-24, -3) | 0.003[2] |
| DBP (mmHg) | -5 (-9, 1) | -5 (-13, 1) | 0.913[2] |
| Total cholesterol (mg/dl) | -4.00 ± 27.87 | -38.84 ± 27.47 | <0.001[1] |
| Triglyceride (mg/dl) | 4 (-9, 33) | -9 (-43, 0) | 0.011[2] |
| HDL-C (mg/dl) | 0 (-5, 4) | 4 (1, 7) | 0.007[2] |
| LDL-C (mg/dl) | -4.96 ± 22.02 | -39.97 ± 26.83 | <0.001[1] |
| FBG (mg/dl) | -3.19 ± 10.03 | -0.87 ± 12.43 | 0.443[1] |
| HbA1c (mg%) | 0.00 (0.20, 0.10) | -0.10 (-0.20, 0.00) | 0.086[2] |
| Creatinine (mg/dl) | 0.08 (-0.01, 0.11) | 0.05 (0.00, 0.13) | 0.981[2] |
| eGRF (ml/min/1.73$^2$) | -6.50 ± 12.65 | -6.40 ± 11.25 | 0.975[1] |
| AST (IU/L) | 2 (-3, 5) | 0 (-3, 3) | 0.250[2] |
| ALT (IU/L) | 0 (-3, 6) | 1 (-4, 5) | 0.827[2] |

[1] Independent t-test (mean ± SD)

[2] Mann-Whitney U test (median [interquartile range]); ALT, alanine aminotransferase; AST, aspartate aminotransferase; BMI, body mass index; DBP, diastolic blood pressure; FBG, fasting blood glucose; HbA1c, hemoglobin A1c; HDL-C, high-density lipoprotein cholesterol; IU/L, international units per liter; kg, kilogram; LDL-C, low-density lipoprotein cholesterol; mmHg, millimeters of mercury; mg, milligrams; mg/dl, milligrams per deciliter; SBP, systolic blood pressure

(4.09 ± 2.70 kg), BMI (1.52 ± 0.92 kg/m$^2$), waist circumference (2.12 ± 2.03 cm), systolic blood pressure (11.65 ± 13.77 mmHg), and total cholesterol (38.84 ± 27.47 mg/dl) (S4 Table).

At the last visit, all participants, including those in the control and intervention groups, as well as those in the bloating, gas production, stomach pain, diarrhea, and constipation ones, were asked about their experiences. The results showed that there were no notable differences between the probiotics and placebo groups (S5 Table).

## The diversity of the gut microbiota

The alpha diversity analysis was used to examine diversity in the two participant groups before and after intervention by measuring the Shannon diversity (Fig 2A and 2B). In both the probiotics and placebo groups, the alpha diversity index showed higher diversity in the post-intervention with no statistical significance.

The beta diversity was used to determine the similarities and dissimilarities in the composition structure of the group of microbial communities with the weighted UniFrac distances. The principal coordinates analysis (PCoA) illustrated a statistically significant difference (PERMANOVA, p-value = 0.002, q-value = 0.01) in the clustering of fecal samples between the gut microbiomes in both groups pre- and post-intervention (Fig 2C and 2D). The PCoA plots indicated that the probiotics group showed a higher dissimilarity between the pre-intervention and post-intervention results due to a more significant proportion of variation explained in the first and second axes.

## Assessment of the bacterial taxonomic composition

At the phylum level, among the two groups, the most common bacterial phyla were Firmicutes, Bacteroidetes, Actinobacteria, and Proteobacteria. The relative abundance of

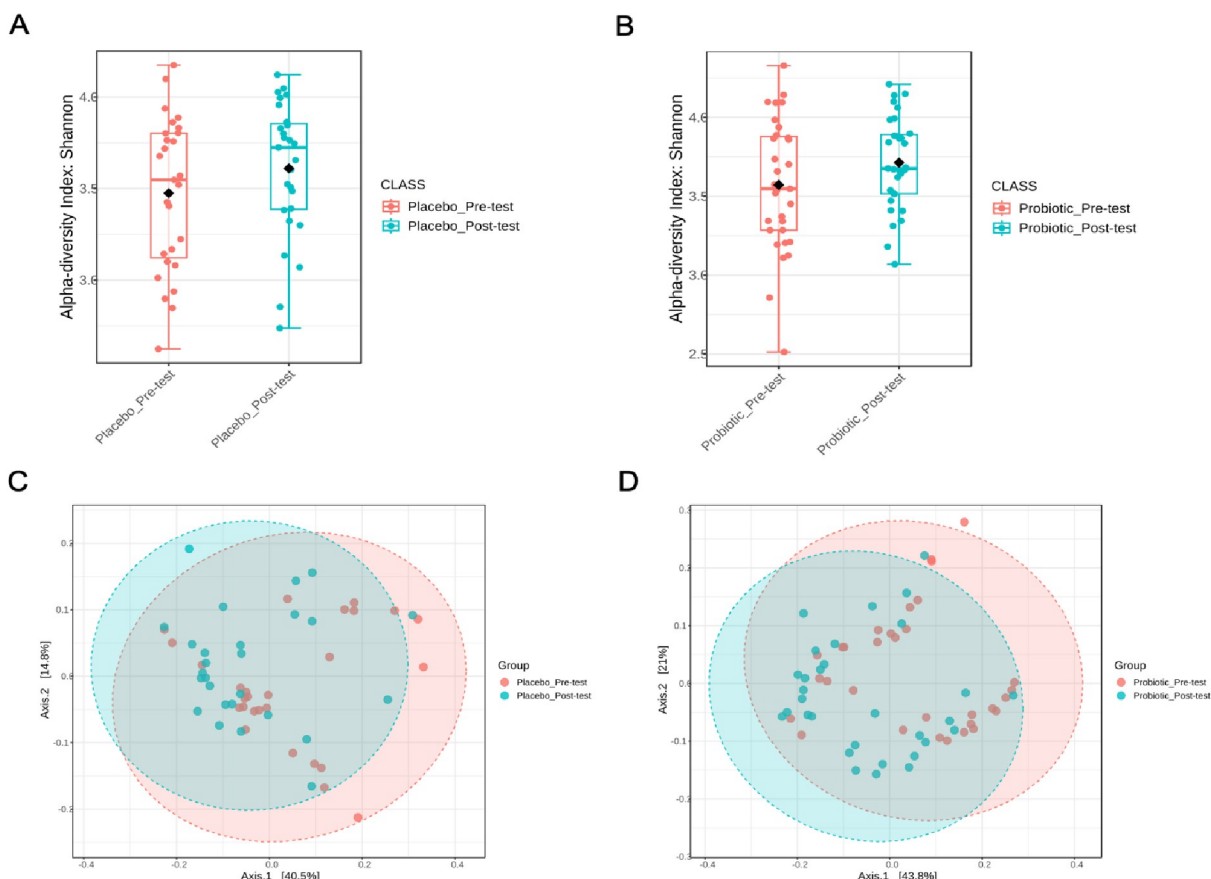

**Fig 2. Alpha- and beta-diversity of microbial composition pre- and post-intervention in both groups.** The Shannon diversity index in placebo (A) and probiotics (B), and principal coordinate analysis (PCoA) of beta diversity based on weighted UniFrac distances in placebo (C) and probiotics (D).

Proteobacteria decreased in the post-intervention probiotics group compared to the pre-intervention one, while in the placebo group, there was no difference (the proportion of Proteobacteria in the post-intervention probiotics group, the pre-intervention probiotics group, the post-intervention placebo group, and the pre-intervention placebo group: 2.10%, 3.02%, 3.53%, and 3.77%, respectively) (S1A and S1B Fig).

At the family level, the Lachnospiraceae and Bacteroidaceae families were shown to be the two most common families. Compared to the pre-intervention probiotics group, the Prevotellaceae and Enterobacteriaceae families were less prevalent proportionately in the post-intervention probiotics group (the proportion of Prevotellaceae and Enterobacteriaceae families in the post-intervention probiotics group and the pre-intervention probiotics group: 10.19% vs. 13.43% and 0.98% vs. 1.50%, respectively) (S1C and S1D Fig).

At the genus level, the highest abundance in each group belonged to the genus *Bacteroides*. In the post-intervention probiotics group, the *Bacteroides*, *Prevotella*, and *Megamonas* genera were less prevalent than in the pre-intervention probiotics group (the proportion of *Bacteroides*, *Prevotella*, and *Megamonas* genera in the post-intervention probiotics group and the pre-intervention probiotics group: 14.41% vs 20.99%, 10.01% vs 13.07%, and 2.48% vs 3.63%, respectively). Moreover, the relative abundance of *Blautia*, *Roseburia*, and *Ruminococcus* genera was greater in the post-intervention probiotics group than in the other (the proportion of *Blautia*, *Roseburia*, and *Ruminococcus* genera in the post-intervention probiotics group and

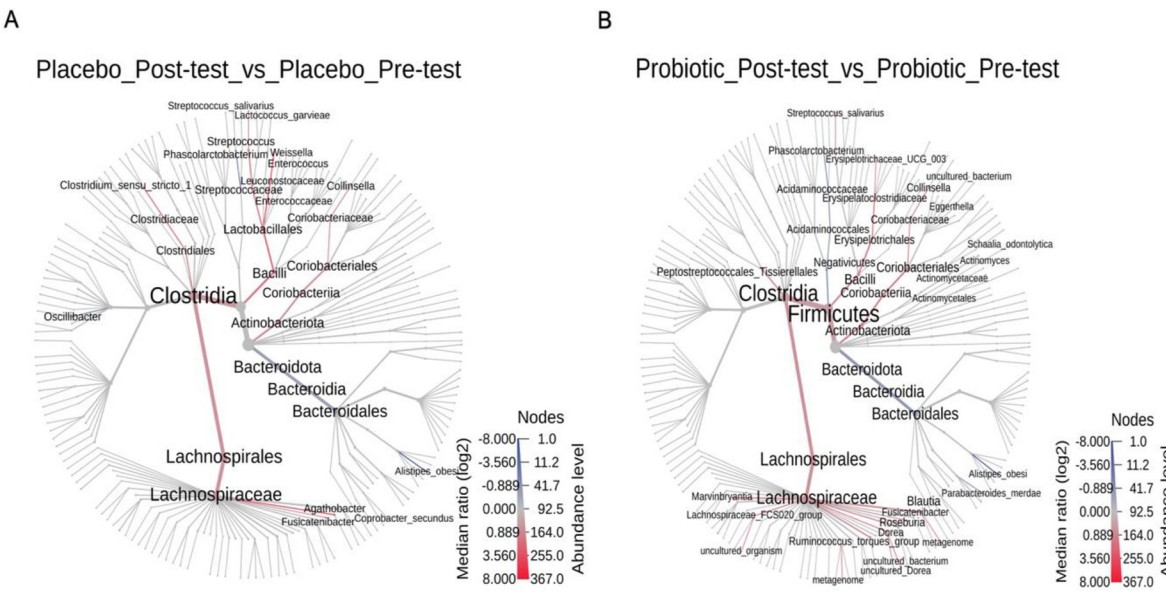

**Fig 3. Heat tree analysis.** Comparison of bacterial microbiota pre-intervention and post-intervention in placebo (A) and probiotics (B).

the pre-intervention probiotics group: 10.22% vs. 6.16%, 2.52% vs. 1.62%, and 0.89% vs. 0.85%, respectively) (S1E and S1F Fig).

According to the pairwise phylogenetic heat tree, the abundance of *Blautia*, *Roseburia*, *Collinsella*, and *Ruminococcus* was greater in the post-intervention probiotics group than in the other group, with statistical significance. Conversely, there was a significantly lower detection of *Parabacteroides merdae* and *Alistipes obesi* in this group. However, there was no change in the placebo one (Fig 3A and 3B).

## Assessment of the predicted function of the gut microbiome

The predicted functional changes were based on the differential abundance pre- and post-intervention in the placebo and probiotics groups. Remarkably, during the 12 weeks, the post-intervention probiotics group showed significant pathway enrichment in ATP-binding cassette (ABC) transporters, in addition to ribonucleic acid (RNA) transport, the biosynthesis of unsaturated fatty acids, glycerophospholipid metabolism, and pyruvate metabolism (Fig 4).

## Discussion

This double-blind, randomized controlled trial investigated the efficacy of *Lacticaseibacillus paracasei* MSMC39-1 and *Bifidobacterium animalis* TA-1 in reducing metabolic syndrome risk factors, particularly LDL-C levels, which are a key risk factor for coronary artery disease and other cardiovascular conditions [33, 49, 50]. Vegetable-based pellets were used as the placebo in both groups and had no effect on the study outcomes. The primary objective was to evaluate the effectiveness of the probiotics in reducing the risk of the characteristics of metabolic syndrome.

The gut microbiota typically requires weeks to months to exhibit measurable changes in response to probiotics. While short-term effects (e.g., changes in metabolite levels) can occur sooner, 12 weeks allows sufficient time for sustained microbial colonization, shifts in microbial diversity, and host responses [51, 52]. Many health benefits linked to probiotics, such as improved gastrointestinal function, immune response, or systemic inflammation, often

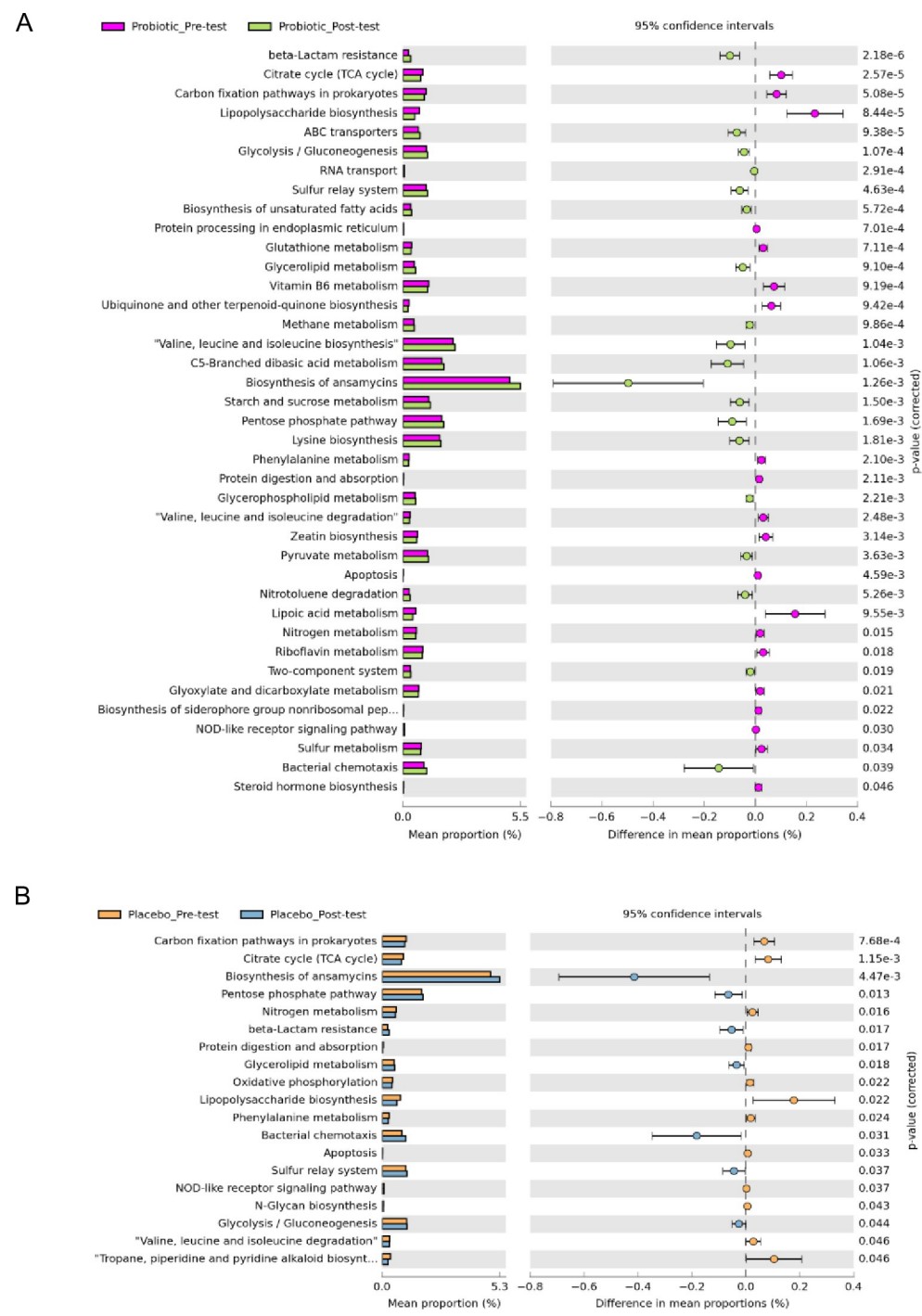

**Fig 4. Functional characterization of different intervention groups based on PICRUSt2 analysis.** Bar chart showing the functional difference (corrected p-value < 0.05) between placebo pre-test and post-test (A), probiotic pre-test and post-test (B).

manifest after 8–12 weeks [53–55]. Therefore, evaluating at 12 weeks ensures that intermediate and longer-term outcomes can be captured reliably. A systematic review and meta-analysis of randomized clinical trials concluded that probiotic supplementation significantly reduces total cholesterol and LDL-C levels, with a stronger effect on LDL-C observed in interventions

lasting longer than six weeks [56]. Likewise, a randomized controlled study found that *L. plantarum* strains (CECT7527, CECT7528, and CECT7529) reduced LDL-C and total cholesterol after 12 weeks compared to placebo, with the LDL-C-lowering effect potentially influenced by individual gut microbiota differences [57]. Similarly, this study conducted a 12-week follow-up to assess laboratory changes, aiming to validate and expand upon these findings. Furthermore, a 12-week intervention allows for observing effects while ensuring participant adherence. Longer durations may increase dropout rates, weakening the study. This timeframe helps reduce transient variations or short-term influences (e.g., dietary changes, seasonal effects) that could confound results.

In this investigation, the LDL-C levels in the intervention group decreased significantly after receiving the probiotics compared to the placebo one. High levels of LDL-C are a risk factor for diabetes mellitus and cardiovascular disease [49, 58–60]. It has been reported that a 1% reduction in blood lipid levels is associated with a 2%–3% reduction in the risk of coronary artery disease [61]. Previous works have reported that both strains of probiotics can reduce blood glucose and lipid levels in both *in vitro* and animal models. Moreover, *L. paracasei* MSMC39-1 can reduce inflammation, and *B. animalis* TA-1 can create bile salt hydrolase (BSH), which reduces cholesterol [28, 62]. Studies have reported that several mechanisms in probiotics can reduce cholesterol levels. For example, probiotics can use cholesterol to form cell membranes, increasing their strength. This helps to reduce the amount of cholesterol in food passages [63, 64]. Probiotics can also convert cholesterol to coprostanol in the intestines and excrete it directly in the feces. This reduces the amount of cholesterol that is absorbed in the intestines and results in a reduction of cholesterol accumulation in the body [34]. The breakdown of bile salts in the digestive tract is attributed to BSH, which is another way in which probiotics can decrease cholesterol. It is then eliminated through the feces and is in an insoluble state. Consequently, the liver must extract more cholesterol to produce bile salts. This is an indirect method of decreasing cholesterol [65, 66].

In this research, the total cholesterol and triglyceride levels decreased significantly after the consumption of the probiotics for 12 weeks. Compared with pre-intervention placebo values, total cholesterol declined by 16.05% in the probiotic group versus 1.67% in placebo, while triglycerides fell by 7.04% versus 3.54%, respectively. Although HDL-C decreased in the probiotic group, the change was not statistically significant. These results align with previous trials: one involving dyslipidemic patients showed a 13.6% reduction in total cholesterol in the *L. plantarum* group after 12 weeks [67], and another by Keleszade et al. found that six weeks of *L. plantarum* ECGC 13110402 use significantly lowered total cholesterol and LDL-C levels [68]. Additionally, *B. longum* CCFM1077 reduced total cholesterol by 13.87% in hyperlipidemic patients, and synbiotic foods containing *L. sporogenes* and inulin improved triglycerides and HDL-C [69]. Overall, both fermented milk products and probiotics significantly reduced total cholesterol and LDL-C, with interventions exceeding four weeks proving more effective [70].

Probiotics could influence SBP through several mechanisms related to gut microbiome modulation [71]. These outcomes are a result of the complex interactions that occur between the host's physiological processes, their metabolic products, and the gut microbiome [72]. Certain probiotic strains can ferment and create peptides that mimic the effects of angiotensin-converting enzyme inhibitors, a type of medication used to treat hypertension [73]. These peptides have the ability to block the angiotensin-converting enzyme, which lowers the amount of angiotensin II, a strong vasoconstrictor, produced [73]. The body's levels of pro- and anti-inflammatory cytokines may be affected by probiotics. Probiotics could improve endothelial function and reduce arterial stiffness, essential for maintaining normal blood pressure, by lowering inflammation, especially in the vascular system [53]. The gut-brain axis, or biochemical communication between the central nervous system and the gastrointestinal tract, may also be

affected by probiotics. In addition, blood pressure may be impacted by this interaction because the autonomic nervous system regulates blood vessel constriction and the heart rate [74]. These clinical trials have investigated the impact of probiotics on blood pressure, with some reporting modest reductions in SBP reductions (reduced SBP by 3.05 mmHg and DBP by 1.51 mmHg), which is consistent with our work (reduced SBP by 10 mmHg with statistical significance, DBP 5 mmHg with no statistical significance) [75, 76]. The extent of blood pressure reduction often depends on the baseline levels, the strains of probiotics utilized, and the duration of the intervention.

In this study, the LDL-C levels showed a marked decrease in the probiotic group compared to the placebo group, dropping by 23.37% and 2.94%, respectively, following the intervention. These results are supported by a meta-analysis of randomized controlled trials, which reported that consuming probiotic *Lactobacillus* strains, particularly *L. reuteri* and *L. plantarum*, significantly reduced both total cholesterol and LDL-C levels [69]. In a controlled, randomized, double-blind trial conducted among dyslipidemic patients, supplementation with a combination of three *L. plantarum* strains (CECT 7527, CECT 7528, and CECT 7529) resulted in a 17.6% reduction in LDL-C levels after 12 weeks compared to the placebo group [67]. In another randomized trial, the administration of *Bifidobacterium longum* CCFM1077 to hyperlipidemic patients reduced LDL-C levels by approximately 13.88%. This benefit may be attributed to the proliferation of anti-obesity-related bacterial genera and favorable shifts in fecal metabolite profiles [77]. In this investigation, we demonstrated that the lipid profile, which includes total cholesterol, LDL-C, HDL-C, and triglycerides, improved following the use of probiotics compared to before. Furthermore, in comparison to the pre-intervention probiotics group, the post-intervention one demonstrated a substantial increase in the relative abundance of *Blautia*, *Collinsella*, *Bifidobacterium*, and *Roseburia*. These gut microorganisms are the short-chain fatty acids (SCFAs)-producing genera. Moreover, higher quantities of SCFAs can be produced more effectively by a more diversified gut microbiota [78, 79].

The fermentation of complex carbohydrates produces metabolites known as SCFAs [80, 81]. Although butyrate can only be generated by members of the phylum Firmicutes, members of the Bacteroidetes phylum can also create acetate [82, 83]. *Eubacterium rectale* and *Roseburia* show a positive connection with SCFAs [84]. Colonocytes and liver gluconeogenesis depend on SCFAs as a primary source of energy. Furthermore, they provide health benefits by strengthening the host's immune response, maintaining the integrity of the intestinal barrier by regulating the expression of tight junction proteins, lowering blood lipid levels by inhibiting the synthesis of cholesterol, and controlling insulin sensitivity [85, 86]. Up to 10% of the host's daily caloric needs may be met using SCFAs as an energy source. For the host to remain healthy and in good condition, these acids, primarily acetic, butyric, and propionic acids, must be present in sufficient amounts in the body [87].

SCFAs can be produced by lactic acid bacteria, such as those of the genera *Lactobacillus* and *Bifidobacterium*, even though these are not classified under the SCFAs category [88]. However, due to the existence of some bacterial species, such as *E. hallii*, which may convert lactate into various SCFAs, it does not accumulate in the colon under normal conditions [78]. It is hypothesized that SCFAs mediate the microbiota-gut-brain axis crosstalk [89]. For instance, certain probiotics (such as *B. longum* SP 07/3 or *B. bifidum* MF 20/5) could produce acetic acid, propionic acid, and lactic acid [87].

At baseline, the gut microbiome profiles displayed low diversity, indicating characteristics of gut dysbiosis, which refers to an imbalance in the microbial community within the gut. This imbalance is caused by an overgrowth of opportunistic bacteria and can lead to various health issues, including inflammatory bowel disease (IBD) [90], obesity [91], and metabolic disorders [92]. In the present study, we observed a significantly higher abundance of

*Phascolarctobacterium* and *Alistipes obesi* prior to the intervention of both treatments. *Alistipes obesi* is often found in higher abundance in individuals with gut dysbiosis and its presence is associated with conditions like non-alcoholic steatohepatitis and liver fibrosis, indicating its role in gut health disturbances [93]. Furthermore, *Alistipes obesi* has the ability to create metabolites that alter the immune response and may be involved in the chronic low-grade inflammation that can occur in metabolic syndrome [94]. Shifts in the levels of *Phascolarctobacterium* have been connected to changes in immune responses. For example, in conditions like psoriasis, an increase in *Phascolarctobacterium* has been linked to abnormal immune activity [95]. This indicates that *Phascolarctobacterium* may be have a role in regulating inflammation and immune responses in the gut.

Patients with autoimmune disorders [96], type 2 diabetes [97], and atherosclerotic diseases [98] frequently have abnormally low levels of SCFA-producing gut microbes [99]. This disruption could result from a decrease in the number of gut microbes that cross-feed SCFA-producing bacteria or from an increase in the synthesis of harmful compounds by the host's gastrointestinal tract or other coexisting microbes [100]. For instance, the absence of butyrate-producing bacteria can disrupt the gut barrier, which in turn can facilitate the release of microbial toxins such as LPS that bind to toll-like receptors and cause inflammation. Higher levels of LPS production in the microbiome of CAD patients have been linked to insulin resistance and abdominal fat [101]. The atherosclerotic process may be protected by SCFAs, whereas lipopolysaccharide (LPS) could trigger inflammation and accelerate the onset of atherosclerosis. The proportion of butyric acid-producing bacteria, such as Lachnospiraceae and Ruminococcaceae, declined as CAD progressed, according to research by Liu H. et al. [102]. Therefore, creating and introducing bacterial occupants (or co-inhabitants) that have the ability to cross-feed and increase/reduce the number of SCFA-producing bacteria in the gut could be a viable probiotic strategy to aid in the treatment of several human illnesses [78, 79].

Patients with obesity typically have altered gut microbiota compositions, suggesting that the gut microbiome may be a contributing component to the development of obesity [103, 104]. Probiotics can be used to manipulate microbial populations in order to enhance the integrity of the intestinal barrier and increase the amount of beneficial bacteria, all of which contribute to weight loss. Moreover, the use of probiotics has been linked to improvements in intestinal dysbiosis and obesity in both humans and animals [105, 106]. These data validate the current investigation, which showed that after receiving both probiotic strains for 12 weeks, volunteers' body weight (5.10%), BMI (4.81%), and waist circumference (5.80%) all significantly dropped from baseline and differed from the placebo group.

*Prevotella* and *Megamonas* genera were less common than the other group in the post-intervention probiotics group. The study conducted by Lopez-Montoya et al. validated our findings, demonstrating a strong correlation between *Megasphaera* and *Escherichia-Shigella* and patients diagnosed with atherogenic dyslipidemia, which is characterized by hypertriglyceridemia and low HDL-C [107]. *Megasphaera* dramatically increased in relative abundance and was linked with inadequate physical activity in individuals who were obese or overweight [108].

An increased risk of cardiovascular disease may be closely linked to diabetes mellitus [109]. A changed gut microbiome can lead to oxidative stress-related diseases, as well as obesity, metabolic endotoxemia, B-cell dysfunction, and systemic inflammation [110, 111]. Probiotic therapy may be useful in managing diabetes mellitus because intestinal bacteria produce more SCFAs. Other works demonstrate that probiotics (*B. animalis* subsp. *lactis* BB-12 and *L. acidophilus* La-5) improve glycemic control in patients with type 2 diabetes mellitus, which can reduce FBG and hemoglobin A1c levels by 6.59% and 11.04%, respectively, although these reductions were not statistically significant [112]. In support of the current investigation, the

FBG level decreased significantly following the use of probiotics compared to the baseline, resulting in reductions of FBG and hemoglobin A1c levels by 0.95% and 3.44%, respectively. Most participants had normal fasting glucose and cumulative hemoglobin A1c levels before the study. There was, therefore, no appreciable difference between the probiotic-treated and control groups.

The microbial functional analysis revealed significant increases in ATP-binding cassette (ABC) transporters, as well as ribonucleic acid (RNA) transport, the biosynthesis of unsaturated fatty acids, glycerophospholipid metabolism, and pyruvate metabolism in the post-intervention probiotics group at 12 weeks. It is assumed that ABC transporter proteins facilitate the ATP-dependent translocation of lipids or lipid-related substances, including cholesterol, bile acids, phospholipids, and sphingolipids [113]. A solute-binding protein-dependent ABC transporter mediates the gut's response of *B. longum* subsp. longum 105-A to the lactulose intake. This reaction contributes to predicting the probiotic responder and non-responder status and offers insight into clinical interventions that evaluate the effectiveness of probiotics [114].

One of the key components of lipid metabolism is RNA transport. The tissue-specific targeting of RNA-based therapies is now possible, increasing the safety of this treatment [115, 116].RNA-based therapies have the potential to dramatically improve outcomes for individuals who have or are at risk for atherosclerotic cardiovascular disease, and multiple putative target proteins have been identified [117]. The biosynthesis of unsaturated fatty acids, which are considered healthy fats, can lower blood cholesterol levels and decrease the risks of cardiovascular diseases [118, 119]. They can be separated into two groups: monounsaturated fats and polyunsaturated fats. Certain unsaturated fatty acids and SCFAs protect against dysbiosis and barrier function impairment. By reducing inflammation brought on by dysbiosis and metabolic endotoxemia, these lipids may increase insulin sensitivity [119]. In cells, glycerophospholipid metabolism plays a role in both metabolism and signaling; it either produces membrane-derived second messengers or is a precursor to them [120]. By affecting colonic glycerophospholipid metabolism and fecal metabolites, the gut microbiota may play a role in the pathophysiology of depression [121]. In addition, one of the key metabolic processes of SCFAs is pyruvate metabolism. Pyruvate, the last product of glycolysis, is crucial for cell metabolism [81, 122].

This study was subject to some limitations, one of which was its use of shotgun sequencing, which produced more information than 16S rRNA V4 region sequencing because of its ability to characterize metabolic pathways and larger gene lists. In addition, potential confounders included baseline variability in diet, physical activity levels, genetic predispositions, and microbiome composition, which may have influenced individual responses to the probiotic intervention. To mitigate adherence-related issues, biweekly follow-ups were conducted during the 12-week intervention to ensure participants achieved at least 80% compliance with the probiotics or placebo regimen. Moreover, despite the lack of standardization in the quality-of-life questionnaire utilized in this trial, the results showed that the intervention was well tolerated and had no negative side effects.

## Conclusion

In summary, this study showed that *Lacticaseibacillus paracasei* MSMC39-1 and *Bifidobacterium animalis* TA-1 probiotics effectively reduced metabolic syndrome risk factors, including significant decreases in low-density lipoprotein cholesterol, triglycerides, and total cholesterol. Improvements in body weight, body mass index, and waist circumference were observed exclusively in the probiotics group. Gut microbiome analysis revealed enhanced beta-diversity and increased abundance of short-chain fatty acid-producing bacteria (*Blautia*, *Roseburia*,

*Collinsella*, and *Ruminococcus*), along with predicted functional changes in ribonucleic acid transport, ATP-binding cassette transporters, and lipid metabolism pathways, supporting the probiotics' role in metabolic regulation.

## Supporting information

**S1 Fig. Bacterial taxonomic profile.** Relative abundance in the placebo and probiotics groups, pre-intervention and post-intervention, at the phylum (A and B), family (C and D), and genus (E and F) levels, respectively.
(TIF)

**S1 Table. Baseline characteristics of the intervention and control groups.**
(DOCX)

**S2 Table. Clinical and laboratory characteristics of male participants.**
(DOCX)

**S3 Table. Clinical and laboratory characteristics of female participants.**
(DOCX)

**S4 Table. Multivariate regression analysis of clinical and laboratory changes.**
(DOCX)

**S5 Table. Quality of life assessment results.**
(DOCX)

## Author Contributions

**Conceptualization:** Wongsakorn Luangphiphat, Pinidphon Prombutara, Krittapat Fukfon, Malai Taweechotipatr.

**Data curation:** Wongsakorn Luangphiphat, Praewpannarai Jamjuree, Chantanapa Chantar-angkul, Porntipha Vitheejongjaroen, Manasvin Onwan.

**Formal analysis:** Wongsakorn Luangphiphat, Pinidphon Prombutara, Praewpannarai Jamjuree, Chantaluck Muennarong, Krittapat Fukfon, Manasvin Onwan, Malai Taweechotipatr.

**Funding acquisition:** Malai Taweechotipatr.

**Investigation:** Wongsakorn Luangphiphat, Praewpannarai Jamjuree, Chantanapa Chantar-angkul, Porntipha Vitheejongjaroen, Chantaluck Muennarong, Malai Taweechotipatr.

**Methodology:** Wongsakorn Luangphiphat, Pinidphon Prombutara, Praewpannarai Jamjuree, Chantaluck Muennarong, Krittapat Fukfon, Manasvin Onwan, Malai Taweechotipatr.

**Project administration:** Malai Taweechotipatr.

**Resources:** Wongsakorn Luangphiphat, Malai Taweechotipatr.

**Software:** Wongsakorn Luangphiphat, Pinidphon Prombutara.

**Supervision:** Pinidphon Prombutara, Malai Taweechotipatr.

**Validation:** Wongsakorn Luangphiphat, Pinidphon Prombutara, Malai Taweechotipatr.

**Visualization:** Wongsakorn Luangphiphat, Pinidphon Prombutara.

**Writing – original draft:** Wongsakorn Luangphiphat, Malai Taweechotipatr.

**Writing – review & editing:** Wongsakorn Luangphiphat, Pinidphon Prombutara, Praewpannarai Jamjuree, Chantanapa Chantarangkul, Malai Taweechotipatr.

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
