## [Decision Letter · Decision Letter 0]

6 Sep 2024

PONE-D-24-31720Efficacy of  Lacticaseibacillus paracasei  MSMC39-1 and Bifidobacterium animalis TA-1 probiotics to modulate gut microbiota and reduce risk factors of metabolic syndrome: A randomized, double-blinded, placebo-controlled studyPLOS ONE

Dear Dr. Taweechotipatr,

Thank you for submitting your manuscript to PLOS ONE. After careful consideration, we feel that it has merit but does not fully meet PLOS ONE’s publication criteria as it currently stands. Therefore, we invite you to submit a revised version of the manuscript that addresses the points raised during the review process.

We look forward to receiving your revised manuscript.

Kind regards,

António Machado

Academic Editor

PLOS ONE

Journal Requirements:

'the Chiangmai Bioveggie and National Innovation Agency (grant number PE0201-02-64-11-0252) and the Center of Excellent in Probiotics Srinakharinwirot University (grant number 324/2565)'

Please state what role the funders took in the study.  If the funders had no role, please state: ''The funders had no role in study design, data collection and analysis, decision to publish, or preparation of the manuscript.'' 

'This study was supported by the Chiangmai Bioveggie and National Innovation Agency (grant number PE0201-02-64-11-0252) and the Center of Excellent in Probiotics Srinakharinwirot University (grant number 324/2565). 

Institutional Review Board Statement: The study was conducted following the Declaration of Helsinki and was approved by the Ethics Committee of Srinakharinwirot University research ethics committees (SWUEC-212/2565F, issued 29 September 2022). The registration number for this study is TCTR20230505002.'

'the Chiangmai Bioveggie and National Innovation Agency (grant number PE0201-02-64-11-0252) and the Center of Excellent in Probiotics Srinakharinwirot University (grant number 324/2565)'

'I have read the journal's policy and the authors of this manuscript have the following competing interests: Pinidphon Prombutara is employed by Mod Gut Co., Ltd. The remaining authors declare that the research was conducted in the absence of any commercial or financial relationships that could be construed as a potential conflict of interest.'

Please confirm that this does not alter your adherence to all PLOS ONE policies on sharing data and materials, by including the following statement: ''This does not alter our adherence to  PLOS ONE policies on sharing data and materials.” (as detailed online in our guide for authors http://journals.plos.org/plosone/s/competing-interests).  If there are restrictions on sharing of data and/or materials, please state these. Please note that we cannot proceed with consideration of your article until this information has been declared. 

8. We note that your paper includes detailed descriptions of individual patients/participants. As per the PLOS ONE policy (http://journals.plos.org/plosone/s/submission-guidelines#loc-human-subjects-research) on papers that include identifying, or potentially identifying, information, the individual(s) or parent(s)/guardian(s) must be informed of the terms of the PLOS open-access (CC-BY) license and provide specific permission for publication of these details under the terms of this license. Please download the Consent Form for Publication in a PLOS Journal (http://journals.plos.org/plosone/s/file?id=8ce6/plos-consent-form-english.pdf). The signed consent form should not be submitted with the manuscript, but should be securely filed in the individual's case notes. Please amend the methods section and ethics statement of the manuscript to explicitly state that the patient/participant has provided consent for publication: “The individual in this manuscript has given written informed consent (as outlined in PLOS consent form) to publish these case details”

Additional Editor Comments:

Dear authors,

I am pleased to say that both reviewers enjoyed the manuscript very much and we are excited about the possibility of publishing your work. However, both reviewers recommended several improvements to the original manuscript. Please read carefully both reviewers’ reports addressing and answering all comments and suggestions.

So, I kindly invite the authors to realize a thoughtful revision of the submitted manuscript to achieve publication endorsement by the reviewers.

Thank you and best regards,

António Machado

Reviewers' comments:

Reviewer's Responses to Questions

**Comments to the Author**

1. Is the manuscript technically sound, and do the data support the conclusions?

Reviewer #1: Yes

Reviewer #2: Partly

2. Has the statistical analysis been performed appropriately and rigorously? 

Reviewer #1: Yes

Reviewer #2: Yes

3. Have the authors made all data underlying the findings in their manuscript fully available?

Reviewer #1: Yes

Reviewer #2: Yes

4. Is the manuscript presented in an intelligible fashion and written in standard English?

Reviewer #1: Yes

Reviewer #2: No

5. Review Comments to the Author

Reviewer #1: This manuscript presents the findings of a randomized, double-blinded, placebo-controlled study that investigates the effects of two probiotic strains, Lacticaseibacillus paracasei MSMC39-1 and Bifidobacterium animalis TA-1, on gut microbiota modulation and the reduction of metabolic syndrome risk factors. The study appears to be well-structured and addresses an important topic within the field of clinical nutrition and metabolic disease management.

The strenghts include:

1. Study Design: The randomized, double-blind, placebo-controlled design is a significant strength of this study. It minimizes bias and strengthens the validity of the results.

2. Clinical Relevance: The study addresses a critical issue—metabolic syndrome—which has significant implications for public health. By focusing on probiotics as a potential intervention, the study taps into a growing interest in non-pharmaceutical approaches to managing chronic conditions.

3. Comprehensive Analysis: The manuscript provides a thorough analysis of both primary and secondary outcomes, including lipid profiles, body composition, and gut microbiome diversity. The use of high-throughput sequencing to analyze gut microbiota is a notable methodological strength

I have the following suggestions to be made:

1. make an adjustment for false discovery rate (FDR) using f.i. the Benjamini-Hochberg method

2. investigate whether gut microbiota (baselina and endpoint) can modulate treatment-related changes in clinical and biochemical features during the study

3. In the discussion use evidence form metaanalyses as these are "strognest" as per EBM

4. discuss all results form the stusy; also the fact that SBP was higher (p<0.05) at baseline in PRO group

Reviewer #2: Title: Efficacy of Lacticaseibacillus paracasei MSMC39-1 and Bifidobacterium animalis TA-1 probiotics to modulate gut microbiota and reduce risk factors of metabolic syndrome: A randomized, double-blinded, placebo-controlled study Autours conducted a clinical study to understand the effect of supplementation of Lacticaseibacillus paracasei MSMC39-1 and Bifidobacterium animalis TA-1 probiotics on characteristics of metabolic syndrome and on composition of Microbiota in adults. The topic is very interesting. However, the writing style requires a more professional tone. I strongly recommend a thorough review by a native English speaker.

Concerns and comments:

1- Title: Instead of “reduce risk factors of metabolic syndrome” it can be “reduce the risk of charachteristics of metabolic syndrome”

2- The introduction is difficult to follow: needs more organization. Adding subheadings such as “The effect of probiotics on body weight and composition”, “The effect of probiotics on lipid metabolism”, “The effect of probiotics on glucose metabolism” etc.

3- In page 4: “The dyslipidemia and insulin sensitivity in obese mice improved equally when the supplements were given. This improved the animals' metabolic dysfunction”. Please explain the meaning of “:equally”

Study Design:

- Please justify including both genders in this study. Considering the difference in metabolism between males and females on one hand and females in peri- vs. post-menopausal status (age range 18-60), makes it difficult to interpret the results.

Suggestion: Include the separate analysis for males and females in addition to current analysis.

- In inclusion criteria, the citation (5) is not consistent with the way that authors designed the inclusion/exclusion criteria. Please justify the current method utilized here.

Here is the methodology from citation 5:

“MetS was defined by the NCEP ATP III-2005 criterion [15], that is, a person who has 3 or more of the following criteria: (1) elevated waist circumference (EWC): waist circumference ≥102 cm in men and ≥88 cm in women; (2) elevated blood pressure: blood pressure ≥130/85 mm Hg or drug treatment of previously diagnosed hypertension; (3) reduced HDL-C: <40 mg/dL in men and <50 mg/dL in women or specific treatment for reduced HDL-C; (4) elevated TGs: TG level ≥150 mg/dL or drug treatment for elevated TG; and (5) elevated fasting glucose: fasting glucose level of ≥100 mg/L or drug treatment for elevated glucose and previously diagnosed type 2 diabetes.”

- The regiment for probiotic is not described: frequency, and timing (within or between meals) is unclear.

- What is the nature/contact of the placebo?

- Please justify the length of the study (12 weeks)

Results:

Table 1 is not representing the results. It can be relocated to the end of the study design section.

The presentation of the results in table 2 is incomplete and unclear.

1- Please add columns to present the absolute values of the variables.

2- Please add columns to present the baseline and Week 12 values for each group.

3- The direction of current values as change is confusing. Please use negative for the reduction and positive for the increase. Current data for example shows reduction in LDL in placebo group assumingly compared with probiotic group and a reduction in HDL in probiotic group compared with the placebo group which is not consistent with the text.

6. PLOS authors have the option to publish the peer review history of their article (what does this mean?). If published, this will include your full peer review and any attached files.

Reviewer #1: **Yes: **Karolina Skonieczna-Żydecka

Reviewer #2: No

---

## [Author Response · Author response to Decision Letter 0]

13 Sep 2024

Response to Reviewer 1 Comments

We are particularly thankful for your suggestions on methodology, clarification on statistical analysis, and knowledge of this study, which helped us refine our manuscript. Your expertise has guided the improvement of this manuscript and contributed significantly to our development as researchers/writers.

We have carefully considered all your comments and have made the necessary revisions to address the issues you highlighted as follows. 

Comments from reviewer

Comment 1: Make an adjustment for false discovery rate (FDR) using f.i. the Benjamini-Hochberg method

Response 1: We have made the statistical adjustment for the false discovery rate (FDR) using the Benjamini-Hochberg. The PERMANOVA and Wilcoxon rank-sum test was followed by a Benjamini-Hochberg (FDR) correction for multiple tests. We have also revised the method in the statistical analysis and visualization session to include the corrections made to the p-value. The authors have revised the first paragraph of statistical analysis and visualization on page 7 to include the relevant text as follows.

“Stata/SE 16.1 software (StataCorp LP, College Station, TX, USA) was used to analyze the statistical data. Statistics were considered significant when p-value < 0.05. All study variables were subjected to descriptive statistics analysis, which was provided as frequency (%) for categorical data and mean ± standard deviation (SD) or median for nonnormal quantitative data. Independent t-test with p-value < 0.05 was utilized if the distribution of the quantitative data, such as age and laboratory results, was normal. The Mann-Whitney U test with p-value <0.05 was used if the data distribution was not normal. The Chi-square test or Fisher's exact test was used to compare the two groups for categorical data. Benjamini-Hochberg (FDR) correction was applied to p-values for multiple hypothesis testing.”

Comment 2: Investigate whether gut microbiota (baseline and endpoint) can modulate treatment-related changes in clinical and biochemical features during the study

Response 2: We conducted a comprehensive investigation into how gut microbiota (baseline and endpoint) modulate treatment-related clinical and biochemical feature changes. This thorough examination allowed us to revise the discussion accordingly, ensuring that your comment was taken seriously and acted upon diligently. The authors have revised the sixth paragraph of the discussion on page 12 to include the relevant text as follows.

“Recent studies have shown that gut microbiota can indeed modulate the effects of probiotic treatments on lipid profiles, influencing both clinical and biochemical features. For example, some probiotics can increase the gut microbiome's ability to produce short-chain fatty acids (SCFAs), which can promote lipid metabolism and lower LDL-C levels. These modifications were accompanied by modifications in the makeup of the gut microbiome, indicating a clear connection between improvements in the lipid profile and gut microbiota regulation (35). In this study, we demonstrated that the lipid profile, which includes total cholesterol, triglycerides, LDL-C, and HDL-C, improved following the probiotic use as compared to before. Furthermore, compared to the pre-intervention probiotics group, the post-intervention probiotics group showed a substantial increase in the relative abundance of Blautia, Collinsella, Bifidobacterium, and Roseburia. These gut microorganisms are the short-chain fatty acids (SCFAs)-producing genera. Moreover, higher quantities of SCFAs can be produced more effectively by a more diversified gut microbiota (36).”

The authors have revised the ninth paragraph of the discussion on page 13 to include the relevant text as follows.

“At baseline, the gut microbiome profiles displayed low diversity, indicating characteristics of gut dysbiosis, which refers to an imbalance in the microbial community within the gut. This imbalance is caused by an overgrowth of opportunistic bacteria and can lead to various health issues, including inflammatory bowel disease (IBD), obesity, and metabolic disorders. In the present study, we observed a significantly higher abundance of Phascolarctobacterium and Alistipes obesi prior to the intervention of both treatments. Alistipes obesi is often found in higher abundance in individuals with gut dysbiosis and its presence is associated with conditions like non-alcoholic steatohepatitis and liver fibrosis, indicating its role in gut health disturbances (41). Furthermore, Alistipes obesi has the ability to create metabolites that alter the immune response and may be involved in the metabolic syndrome's chronic low-grade inflammation (42). Changes in the levels of Phascolarctobacterium have been connected to changes in immune responses. For example, in conditions like psoriasis, an increase in Phascolarctobacterium has been linked to abnormal immune activity (43). This indicates that Phascolarctobacterium may be involved in regulating inflammation and immune responses in the gut.”

Comment 3: In the discussion use evidence from meta-analyses as these are "strongest" as per EBM

Response 3: We completely agree that clinical decisions should be based on the totality of evidence, not just on a single study, and meta-analyses offer the best available evidence for decision-making in clinical practice. Therefore, we have revised the discussion to include more meta-analyses in the fourth paragraph of the discussion on pages 12-13 as follows. 

“In addition, a meta-analysis of randomized controlled trials found that consuming probiotic Lactobacillus, particularly strains like L. reuteri and L. plantarum, significantly reduced total cholesterol and LDL-C levels. The study also noted beneficial effects on triglycerides and HDL-C when consuming synbiotic foods containing L. sporogenes and inulin (30). Another meta-analysis showed that probiotic interventions, including fermented milk products and probiotics, significantly reduced total cholesterol and LDL-C levels. Long-term probiotic interventions (more than 4 weeks) were more effective in decreasing these lipid levels than short-term interventions (31).” 

Comment 4: Discuss all results from the study; also the fact that SBP was higher (p<0.05) at baseline in the probiotic group

Response 4: We have revised the discussion to cover all results, including the differences in SBP. The authors have revised the fifth paragraph of the discussion on page 12 to include the relevant text as follows.

“In this study, probiotics can influence SBP through several mechanisms related to gut microbiome modulation. These outcomes are a result of the complex interactions that occur between the host's physiological processes, the gut microbiome, and their metabolic products. Certain probiotic strains can ferment and create peptides that mimic the effects of angiotensin-converting enzyme inhibitors, a type of medication used to treat hypertension. These peptides have the ability to block the angiotensin-converting enzyme, which lowers the amount of angiotensin II, a strong vasoconstrictor, produced (32). The body's levels of cytokines that are pro-inflammatory and anti-inflammatory can be affected by probiotics. Probiotics contribute to improving endothelial function and reducing arterial stiffness, which is essential for maintaining normal blood pressure, by lowering inflammation, especially in the vascular system (33). The gut-brain axis, or the biochemical communication between the central nervous system and the gastrointestinal tract, may also be impacted by probiotics. Blood pressure may be impacted by this interaction because the autonomic nervous system regulates blood vessel constriction and heart rate (34). However, these clinical trials have investigated the impact of probiotics on blood pressure, with some reporting modest reductions in SBP, which is consistent with our study. The extent of blood pressure reduction often depends on the baseline blood pressure levels, the strains of probiotics used, and the duration of the intervention.”

Please know that your feedback is greatly appreciated and has been instrumental in advancing this piece of work. We look forward to working together again and hope the revisions meet your approval.

Again, thank you for your valuable input and for supporting the peer review process.

*********

Response to Reviewer 2 Comments

The authors would like to express our sincere gratitude for your valuable suggestions regarding the title, introduction, data presentation, clarification of the regimen, and overall understanding of this study. Your expertise has greatly contributed to refining and enhancing our manuscript. Moreover, your guidance has been instrumental in our growth and development as researchers and writers. Thank you once again for your insightful feedback and support.

We have carefully considered all your comments and have made the necessary revisions to address the issues you highlighted as follows. 

Comments from reviewer

Comment 1: However, the writing style requires a more professional tone. I strongly recommend a thorough review by a native English speaker.

Response 1: The English language throughout the manuscript has been revised and corrected in accordance with the reviewer's suggestions.

Comment 2: Title: Instead of “reduce risk factors of metabolic syndrome” it can be “reduce the risk of characteristics of metabolic syndrome”

Response 2: Thank you so much for your valuable suggestions. The authors have revised the title as follows. 

“Efficacy of Lacticaseibacillus paracasei MSMC39-1 and Bifidobacterium animalis TA-1 probiotics to modulate gut microbiota and reduce the risk of characteristics of metabolic syndrome: A randomized, double-blinded, placebo-controlled study”

Comment 3: The introduction is difficult to follow: and needs more organization. Adding subheadings such as “The effect of probiotics on body weight and composition”, “The effect of probiotics on lipid metabolism”, “The effect of probiotics on glucose metabolism” etc.

Response 3: The authors have revised the introduction, adding subheadings before paragraph 4 and paragraph 5 on page 3 of the introduction, and edited the text, as follows;

“The effect of probiotics on body weight and composition

Probiotics have emerged as a potential strategy for managing body weight and metabolic syndrome by influencing the gut microbiome. They are commonly used to maintain or restore a healthy microbial balance. Certain probiotic strains may reduce the risk of metabolic syndrome by modulating inflammation. A meta-analysis has shown that, compared to a placebo, probiotics can significantly reduce body weight, fat mass, and body mass index (BMI), highlighting their role as dietary agents in improving metabolic health (13).

The effect of probiotics on lipid metabolism and glucose metabolism 

Several probiotic species, including Bifidobacterium longum, Lactobacillus acidophilus, and Lactobacillus gasseri, have demonstrated beneficial effects, such as the enhancement of lipid metabolic pathways, upregulation of genes involved in carbohydrate transport and metabolism, and improved intestinal digestion. Additionally, these probiotics have been shown to modulate bile acid metabolism, reduce plasma glucose, cholesterol, lipoproteins, and triglycerides, promote bile salt deconjugation, and contribute to weight loss (14). In one mice study, after a 12-week intervention, Thiennimitr et al. assessed the effects of probiotic L. paracasei HII01 (108 CFU/mL), prebiotic xylo-oligosaccharides (XOS) (10%), and synbiotic on the improvement of gut dysbiosis. Supplementation improved both dyslipidemia and insulin sensitivity in obese mice, leading to better overall metabolic function (15). A randomized placebo-controlled clinical trial studied L. plantarum PBS067, L. acidophilus PBS066, and L. reuteri PBS072 with prebiotics in patients with metabolic syndrome. The results showed they could significantly reduce glucose, lipid, and inflammatory mediators (16).” 

Comment 4: On page 4: “The dyslipidemia and insulin sensitivity in obese mice improved equally when the supplements were given. This improved the animals' metabolic dysfunction”. Please explain the meaning of “: equally”

Response 4: The authors have removed “equally” and revised the text in paragraph 3 on page 3 to provide a clearer and more precise description of the study's results, as follows;

“Several probiotic species, including Bifidobacterium longum, Lactobacillus acidophilus, and Lactobacillus gasseri, have demonstrated beneficial effects, such as the enhancement of lipid metabolic pathways, upregulation of genes involved in carbohydrate transport and metabolism, and improved intestinal digestion. Additionally, these probiotics have been shown to modulate bile acid metabolism, reduce plasma glucose, cholesterol, lipoproteins, and triglycerides, promote bile salt deconjugation, and contribute to weight loss (14). In one mice study, after a 12-week intervention, Thiennimitr et al. assessed the effects of probiotic L. paracasei HII01 (108 CFU/mL), prebiotic xylo-oligosaccharides (XOS) (10%), and synbiotic on the improvement of gut dysbiosis. Supplementation improved both dyslipidemia and insulin sensitivity in obese mice, leading to better overall metabolic function (15). A randomized placebo-controlled clinical trial studied L. plantarum PBS067, L. acidophilus PBS066, and L. reuteri PBS072 with prebiotics in patients with metabolic syndrome. The results showed they could significantly reduce glucose, lipid, and inflammatory mediators (16).”

Comment 5: Study Design: Please justify including both genders in this study. Considering the difference in metabolism between males and females on one hand and females in peri- vs. post-menopausal status (age range 18-60), makes it difficult to interpret the results.

Suggestion: Include a separate analysis for males and females in addition to the current analysis.

Response 5: The authors have incorporated Supplementary Tables S2 and S3 and revised the first paragraph on page 9 to include the relevant text.

Table S2. Clinical and laboratory characteristics of male participants in the probiotics and placebo groups (N=58)

Table S3. Clinical and laboratory characteristics of female participants in the probiotics and placebo groups (N=58)

Following the intervention in the probiotics group, reductions were observed in total cholesterol, triglycerides, LDL-C, and HbA1c levels, along with an increase in HDL-C, when compared to the placebo group in both male and female participants (Tables S2-S3).

Comment 6: Study Design: In inclusion criteria, the citation (5) is not consistent with the way that the authors designed the inclusion/exclusion criteria. Please justify the current method utilized here.

Here is the methodology from citation 5: “MetS was defined by the NCEP ATP III-2005 criterion [15], that is, a person who has 3 or more of the following criteria: (1) elevated waist circumference (EWC): waist circumference ≥102 cm in men and ≥88 cm in women; (2) elevated blood pressure: blood pressure ≥130/85 mm Hg or drug treatment of previously diagnosed hypertension; (3) reduced HDL-C: <40 mg/dL in men and <50 mg/dL in women or specific treatment for reduced HDL-C; (4) elevated TGs: TG level ≥150 mg/dL or drug treatment for elevated TG; and (5) elevated fasting glucose: fasting glucose level of ≥100 mg/L or drug treatment for elevated glucose and previously diagnosed type 2 diabetes.”

Response 6: The authors greatly appreciate your thoughtful suggestion, and as a result, we have revised the inclusion criteria on page 4, as follows;

“Inclusion criteria 

The inclusion criteria for participants in this study required individuals to meet the following conditions: (1) have an age range of 18 to 60 years, and (2) exhibit three or more of the following criteria: (a) elevated waist circumference (≥102 cm in men and ≥88 cm in women), (b) elevated blood pressure (≥130/85 mm Hg), (c) reduced high-density lipoprotein cholesterol (HDL-C) levels (<40 mg/dL in men and <50 mg/dL in women), (d) elevated triglycerides (≥150 mg/dL), and (e) elevated fasting blood glucose (100-125 mg/dL).”

Comment 7: Study Design

---

## [Decision Letter · Decision Letter 1]

16 Oct 2024

PONE-D-24-31720R1Efficacy of Lacticaseibacillus paracasei MSMC39-1 and Bifidobacterium animalis TA-1 probiotics to modulate gut microbiota and reduce the risk of characteristics of metabolic syndrome: A randomized, double-blinded, placebo-controlled studyPLOS ONE Dear Dr. Taweechotipatr,

Thank you for submitting your manuscript to PLOS ONE. After careful consideration, we feel that it has merit but does not fully meet PLOS ONE’s publication criteria as it currently stands. Therefore, we invite you to submit a revised version of the manuscript that addresses the points raised during the review process.

I am pleased to inform you that one reviewer (reviewer 1) just asked for minor revisions the revised manuscript and the other reviewer (reviewer 3) requested some revisions for future publication endorsement. Please carefully answer reviewer 3' concerns and rectify the manuscript following his/her comments.

We look forward to receiving your revised manuscript.

Thank you for choosing PLOS ONE journal and best regards,

António Machado

Academic Editor

Reviewers' comments:

Reviewer's Responses to Questions

**Comments to the Author**

1. If the authors have adequately addressed your comments raised in a previous round of review and you feel that this manuscript is now acceptable for publication, you may indicate that here to bypass the “Comments to the Author” section, enter your conflict of interest statement in the “Confidential to Editor” section, and submit your "Accept" recommendation.

Reviewer #1: All comments have been addressed

Reviewer #3: (No Response)

2. Is the manuscript technically sound, and do the data support the conclusions?

Reviewer #1: Yes

Reviewer #3: Yes

3. Has the statistical analysis been performed appropriately and rigorously? 

Reviewer #1: Yes

Reviewer #3: Yes

4. Have the authors made all data underlying the findings in their manuscript fully available?

Reviewer #1: Yes

Reviewer #3: Yes

5. Is the manuscript presented in an intelligible fashion and written in standard English?

Reviewer #1: Yes

Reviewer #3: Yes

6. Review Comments to the Author

Reviewer #1: Thank you for introducing all amendments. While the manuscript has improved from previous versions, a thorough English review is still needed to ensure fluency and professionalism in tone. There is acknowledgment of higher baseline systolic blood pressure in the probiotics group; however, more detailed discussion of how this might influence the overall results would improve the robustness of the study. and did you use this as a covariate/adjustment in the stat analyses? Certain results, like the percentage changes in various biomarkers, could be clearer. Some data points (e.g., absolute versus percentage changes) are not immediately intuitive for the reader.

Reviewer #3: The present is an interesting RCT

Some issues should be addressed

Introduction should be shortened and made more adherent to clinical potential impact of these fidnings. Sub headings should be removed

Methods: it should be added how placebo was created

Methods: sample size calculation is totally not clear to me, an d usually should be performed on primary outcomes

Methods: to correct for reduced sample size sometime mutlivariate analysis is exploited. did authros exploited it?

7. PLOS authors have the option to publish the peer review history of their article (what does this mean?). If published, this will include your full peer review and any attached files.

Reviewer #1: **Yes: **Karolina Skonieczna-Żydecka

Reviewer #3: **Yes: **Fabrizio D'Ascenzo

---

## [Author Response · Author response to Decision Letter 1]

25 Oct 2024

Response to Reviewer 1 Comments

Title: The efficacy of Lacticaseibacillus paracasei MSMC39-1 and Bifidobacterium animalis TA-1 probiotics in modulating gut microbiota and reducing the risk of the characteristics of metabolic syndrome: A randomized, double-blinded, placebo-controlled study

We express our sincere gratitude for your insightful suggestions regarding the methodology, clarification of the statistical analysis, and overall knowledge of this study. Your expertise has significantly contributed to the refinement of our manuscript and has greatly facilitated our development as researchers and writers.

We have meticulously considered all of your comments and implemented the necessary revisions to address the issues you identified, as outlined below.

Comments from reviewer

Comment 1: English review is still needed to ensure fluency and professionalism in tone.

Response 1: 

Thank you for your kind suggestion. We have submitted our manuscript for professional language proofreading to "English Proofread." Following receipt of the edited document, we thoroughly reviewed and incorporated the recommended corrections throughout the manuscript. This process has significantly improved the overall quality and clarity of the text.

Comment 2: There is acknowledgment of higher baseline systolic blood pressure in the probiotics group; however, a more detailed discussion of how this might influence the overall results would improve the robustness of the study. and did you use this as a covariate/adjustment in the stat analyses? 

Response 2: 

Thank you for your valuable suggestion. We have incorporated a multivariate regression analysis, as presented in Table S4, to account for potential confounding factors, including the higher baseline systolic blood pressure in the probiotics group. These adjustments enhance the robustness of the study findings by addressing the influence of baseline differences. The authors have added the seventh paragraph of the results section (lines 279-285) to include the relevant text as follows. 

“Multivariate regression analysis revealed that the clinical and laboratory characteristics, as well as changes in variables between the probiotics and placebo groups, remained consistent. After 12 weeks of administration of the probiotic strains L. paracasei MSMC39-1 and B. animalis TA-1, a significant reduction in LDL-C after taking probiotics was achieved (the differences were 35.09, p-value <0.001). Additionally, participants in the probiotics group exhibited significant reductions in body weight, BMI, waist circumference, SBP, and total cholesterol compared to the placebo group at baseline and after 12 weeks (Table S4).”

Comment 3: Certain results, like the percentage changes in various biomarkers, could be clearer. Some data points (e.g., absolute versus percentage changes) are not immediately intuitive for the reader.

Response 3: 

Thank you for your helpful feedback. To improve clarity, we have revised the explanation of Table 2 on page 8 (lines 248-249): “The differences between the pre-intervention and post-intervention results in the placebo and probiotics groups at 12 weeks are presented in Table 2." We hope this adjustment makes the data, including percentage and absolute changes, clearer and more intuitive for readers. 

We would like to express our sincere gratitude for your valuable feedback, which has been instrumental in improving and advancing this work. We look forward to the opportunity to collaborate again in the future and trust that the revisions meet your approval.

Once again, thank you for your thoughtful input and your continued support of the peer review process.

*********

Response to Reviewer 3 Comments

Title: The efficacy of Lacticaseibacillus paracasei MSMC39-1 and Bifidobacterium animalis TA-1 probiotics in modulating gut microbiota and reducing the risk of the characteristics of metabolic syndrome: A randomized, double-blinded, placebo-controlled study

We are especially grateful for your valuable suggestions on the methodology, clarification of the statistical analysis, and insightful knowledge regarding this study. Your expertise has greatly contributed to the refinement of our manuscript and has played a pivotal role in our growth as researchers and writers. Your guidance has been instrumental in enhancing the quality of this work.

We have thoroughly reviewed all of your comments and have implemented the necessary revisions to address the concerns you raised, as detailed below.

Comments from reviewer

Comment 1: 

The introduction should be shortened and made more adherent to the clinical potential impact of these findings. Subheadings should be removed

Response 1: 

The introduction has been revised to focus more concisely on the clinical potential impact of the findings, emphasizing their relevance to patient outcomes and therapeutic implications. The unnecessary details have been removed to streamline the narrative and align it with the study's clinical significance. Additionally, all subheadings have been eliminated to ensure a smoother and more cohesive flow throughout the manuscript, as per your request. The authors have revised the second to fourth paragraphs of the introduction on pages 2-4 (lines 56-87) to include the relevant text as follows. 

“The gastrointestinal tract is colonized by the intestinal microbiota, a key environmental factor that directly influences host health and contributes to the exacerbation of various diseases (6). Dysbiosis, an imbalance in the microbiota, reduces its diversity and function, leading to metabolic issues. A high-fat diet can worsen dysbiosis by increasing circulating lipopolysaccharide (LPS) and lipid levels, as well as compromising the intestinal barrier. Additionally, dysbiosis may contribute to chronic inflammation and the development of obesity, diabetes mellitus, and metabolic syndrome through interactions between genetic and environmental factors (7,8). Treatment for metabolic syndrome and obesity-related metabolic endotoxemia involves preventing microbial dysbiosis and maintaining intestinal barrier integrity (9).

According to Hill et al. (10), the consensus statement that the International Scientific Association for Probiotics and Prebiotics (ISAPP) proposed for the proper usage of the term probiotic refers to live microorganisms that, when administered in adequate amounts, confer a health benefit on the host. Commercial strains are more likely to contain probiotics from the genera Lactobacillus and Bifidobacterium (11). The key health benefits of probiotics include immune modulation, enhanced mineral and vitamin absorption, constipation relief, microbiome regulation, post-antibiotic microbiome stabilization, increased gastrointestinal resistance to pathogens, and pathogen reduction through short-chain fatty acid (SCFA) production (12).

Probiotics offer a potential strategy for managing body weight as well as metabolic syndrome by modulating the gut microbiome. They help maintain microbial balance and may reduce the risk of metabolic syndrome by controlling inflammation. A meta-analysis showed that probiotics, compared to placebos, significantly reduce body weight, fat mass, and the body mass index (BMI), supporting their role in improving metabolic health (13). Probiotic species like Bifidobacterium longum, Lactobacillus acidophilus, and Lactobacillus gasseri have demonstrated benefits by enhancing lipid metabolism, upregulating carbohydrate transport and metabolism genes, and improving digestion. They also modulate bile acid metabolism, reduce plasma glucose, cholesterol, and triglycerides, promote bile salt deconjugation, and aid in weight loss (14). A 12-week mice study confirmed that L. paracasei HII01, xylo-oligosaccharides (XOS), and synbiotics improved dyslipidemia and insulin sensitivity, enhancing metabolic functions (15). Similarly, a randomized placebo-controlled trial found L. plantarum PBS067, L. acidophilus PBS066, and L. reuteri PBS072, along with prebiotics significantly reduced glucose, lipids, and inflammatory mediators in metabolic syndrome patients (16).”

Comment 2: Methods: it should be added how the placebo was created

Response 2: 

In response to the feedback, the Methods section has been updated to include details on the creation of the placebo. The authors have revised the first paragraphs of the Methods section on pages 3-4 (lines 105-109) to include the relevant text as follows. 

“The tablets were manufactured at a good manufacturing practice (GMP) certified food production facility. Probiotics with a vegetable-based pellet and a placebo vegetable-based pellet were compressed into tablets using microcrystalline cellulose (MCC) as a binder to enhance cohesion. The tablets were coated with hydroxypropyl methylcellulose (HPMC) to protect against moisture.”

Comment 3: Methods: sample size calculation is totally not clear to me, and usually should be performed on primary outcomes 

Response 3: 

Thank you for your feedback. The sample size calculation has been revised for clarity and is now based on the primary outcome of the study. The authors have revised the first paragraphs of the Methods section, sample size calculation on page 5 (lines 162-168), as follows. 

“The primary outcome was utilized to determine the sample size, based on previous research that investigated LDL-C changes between placebo and intervention groups (17). The placebo group exhibited a change of 0.17 ± 0.46, while the intervention group showed a difference of -0.40 ± 0.70. Using these values, the sample size was calculated with STATA software, setting α = 0.05 and power (1-β) = 0.90. The required sample size was determined to be 24 participants per group. To account for potential missing data (approximately 20%), the sample size was adjusted to approximately 28 participants per group.”

Comment 4: Methods: to correct for reduced sample size sometimes multivariate analysis is exploited. did authors exploit it? 

Response 4: 

In response to the query, multivariate regression analysis was employed to account for the reduced sample size and control for potential confounding factors. The multivariate regression analysis is presented in Table S4 and has been further explained in the Results section on page 9 (lines 279-285) to provide a detailed interpretation of the findings, as follows. 

“Multivariate regression analysis revealed that the clinical and laboratory characteristics, as well as changes in variables between the probiotics and placebo groups, remained consistent. After 12 weeks of administration of the probiotic strains L. paracasei MSMC39-1 and B. animalis TA-1, a significant reduction in LDL-C after taking probiotics was achieved (the differences were 35.09, p-value <0.001). Additionally, participants in the probiotics group exhibited significant reductions in body weight, BMI, waist circumference, SBP, and total cholesterol compared to the placebo group at baseline and after 12 weeks (Table S4).”

Your feedback is sincerely appreciated and has been pivotal in improving this work. We look forward to the opportunity to collaborate again in the future and trust that the revisions align with your expectations.

Once again, we extend our gratitude for your valuable insights and for your support of the peer review process.

---

## [Decision Letter · Decision Letter 2]

13 Dec 2024

PONE-D-24-31720R2The efficacy of Lacticaseibacillus paracasei MSMC39-1 and Bifidobacterium animalis TA-1 probiotics in modulating gut microbiota and reducing the risk of the characteristics of metabolic syndrome: A randomized, double-blinded, placebo-controlled studyPLOS ONE

Dear Dr. Taweechotipatr,

Thank you for submitting your manuscript to PLOS ONE. After careful consideration, we feel that it has merit but does not fully meet PLOS ONE’s publication criteria as it currently stands. Therefore, we invite you to submit a revised version of the manuscript that addresses the points raised during the review process.

We look forward to receiving your revised manuscript.

Kind regards,

António Machado, PhD

Academic Editor

PLOS ONE

Journal Requirements:

**Additional Editor Comments:**

Dear authors,

I am pleased to inform you that one reviewer already accepted the revised manuscript for publication and the other reviewer (reviewer 4) requested some revisions for publication endorsement. Please carefully answer reviewer 4' concerns and rectify the manuscript following his/her comments.

Thank you for choosing PLOS ONE journal and best regards,

António Machado

Reviewers' comments:

Reviewer's Responses to Questions

**Comments to the Author**

1. If the authors have adequately addressed your comments raised in a previous round of review and you feel that this manuscript is now acceptable for publication, you may indicate that here to bypass the “Comments to the Author” section, enter your conflict of interest statement in the “Confidential to Editor” section, and submit your "Accept" recommendation.

Reviewer #3: All comments have been addressed

Reviewer #4: (No Response)

2. Is the manuscript technically sound, and do the data support the conclusions?

Reviewer #3: (No Response)

Reviewer #4: Yes

3. Has the statistical analysis been performed appropriately and rigorously? 

Reviewer #3: (No Response)

Reviewer #4: I Don't Know

4. Have the authors made all data underlying the findings in their manuscript fully available?

Reviewer #3: (No Response)

Reviewer #4: Yes

5. Is the manuscript presented in an intelligible fashion and written in standard English?

Reviewer #3: (No Response)

Reviewer #4: Yes

6. Review Comments to the Author

Reviewer #3: (No Response)

Reviewer #4: The results presented in this study are relevant and important in the field of analysis of the beneficial effects of postbiotics. However, I recommend making the following changes:

- Move the number of participants in the two groups to lines 37 and 38 of the abstract to avoid confusion.

- Place the units of all parameters evaluated and detailed in the text. Example: line 45, the value 39.97 has no units. Revise these errors in the complete text.

- Change the term percentage for its sign (%) in line 73 and revise these errors in the complete text.

- Important statements are made in the introduction and in the discussions. These should be properly cited. There are long and important sentences that are supported with only one bibliographic reference. It is important to add more updated references.

- Explain what technique was used to randomize the study population.

- Put references in lines 176 to 178, 517 to 522 and other paragraphs such as these that do not have bibliographic references.

- Lines 183 to 185 are results, they should not be in methods.

- Justify why it was decided to evaluate the results at 12 weeks.

- Improve the explanation of the sample size calculation. Why LDL-C is chosen for the calculation?

- Make a table summarizing the results explained in lines 411 to 416.

- Correct Figure 4. Represent the placebo and postbiotic groups in the same color in A and B to avoid confusion. Improve the caption of the image. What exactly does A and B represent?

- In discussions, discuss how much cholesterol, triglycerides, glucose, etc. are reduced. Compare with the cited studies by expressing the percentages or values reported to know if the variability is similar or different from the study.

- Focus discussions on the probiotics that were used. Too much is said about other probiotics not related to the species used.

- In the limitations of the study, indicate possible potential confounders.

- Only determine causality in the results obtained when they are absolutely certain that they are exclusively due to the intervention evaluated.

- Conclusions are similar to the abstract. Restructure the conclusions.

7. PLOS authors have the option to publish the peer review history of their article (what does this mean?). If published, this will include your full peer review and any attached files.

Reviewer #3: **Yes: **Fabrizio D'Ascenzo

Reviewer #4: **Yes: **Sandra Pamela Cangui Panchi

---

## [Author Response · Author response to Decision Letter 2]

19 Dec 2024

Response to Reviewer 4 Comments

Title: The efficacy of Lacticaseibacillus paracasei MSMC391 and Bifidobacterium animalis TA1 probiotics in modulating gut microbiota and reducing the risk of the characteristics of metabolic syndrome: A randomized, double-blinded, placebo-controlled study

We sincerely thank you for your thorough and constructive feedback on our manuscript. We deeply appreciate the time and effort dedicated to reviewing our work, and we have carefully considered each comment to revise and improve the manuscript accordingly. Below, we present a detailed, point-by-point response to the reviewers' comments. Revisions have been highlighted in the manuscript for ease of reference.

Reviewer comments and our responses:

1. Move the number of participants in the two groups to lines 37 and 38 of the abstract to avoid confusion. 

 Response: The number of participants in both groups has been moved to lines 37 and 38 in the abstract to enhance clarity, as follows. 

“Fifty-eight participants with risk factors of metabolic syndrome according to the inclusion criteria were randomized into two groups and given probiotics (Lacticaseibacillus paracasei MSMC39-1 and Bifidobacterium animalis TA-1) (n=31) or a placebo (n=27).” (line 27)

2. Place the units of all parameters evaluated and detailed in the text. Example: line 45, the value 39.97 has no units. Revise these errors in the complete text. 

 Response: Units have been added to all parameters throughout the manuscript, including the example on line 45. The text has been thoroughly reviewed to ensure all numerical values are appropriately labeled, as follows.

- “The primary outcome was achieved by the probiotics group as their low-density lipoprotein-cholesterol level dramatically lowered compared to the placebo group (the difference was 39.97 ± 26.83 mg/dl, p-value <0.001).” (line 32)

- “Participants who met the following criteria were excluded: (1) they had been diagnosed with the following underlying diseases: diabetes mellitus (hemoglobin A1c [HbA1c] ≥ 6.5 mg%)” (line 147)

- “Following the administration of both strains of probiotics, namely L. paracasei MSMC39-1 and B. animalis TA-1 at 12 weeks, the difference in participants' body weight, BMI, and waist circumference, all significantly decreased from the baseline (3.90 kg, 1.36 kg/m2, and 2.00 cm, p-value <0.001 for body weight, BMI, and waist circumference, respectively), and the variation differed significantly from the placebo group (Table 2).” (line 256)

- “The total cholesterol, triglycerides, and LDL-C levels, considerably decreased after taking probiotics compared to before with statistical significance (the differences were 38.84 ± 27.47 mg/dl, p-value <0.001; 9 mg/dl, p-value 0.011; and 39.97 ± 26.83 mg/dl, p-value <0.001, respectively).” (line 262)

- “After 12 weeks of administration of the probiotic strains L. paracasei MSMC39-1 and B. animalis TA-1, a significant reduction in LDL-C after taking probiotics (39.97 ± 26.83 mg/dl) was achieved (p-value <0.001). Additionally, compared to the placebo group, participants in the probiotics group showed significant reductions at 12 weeks from baseline in body weight (4.09 ± 2.70 kg), BMI (1.52 ± 0.92 kg/m²), waist circumference (2.12 ± 2.03 cm), systolic blood pressure (11.65 ± 13.77 mmHg), and total cholesterol (38.84 ± 27.47 mg/dl) (Table S4).” (line 287-291)

3. Change the term percentage for its sign (%) in line 73 and revise these errors in the complete text. 

 Response: The term "percentage" has been replaced with the sign (%) in previous line 73 and similar instances throughout the manuscript. 

4. Important statements are made in the introduction and in the discussions. These should be properly cited. There are long and important sentences that are supported with only one bibliographic reference. It is important to add more updated references. 

 Response: Additional updated references have been added to support important statements in both the introduction and discussion sections. These references are appropriately cited in the text, as follows. 

- “This double-blind, randomized controlled trial investigated the efficacy of Lacticaseibacillus paracasei MSMC39-1 and Bifidobacterium animalis TA-1 in reducing metabolic syndrome risk factors, particularly LDL-C levels, which are a key risk factor for coronary artery disease and other cardiovascular conditions (33,49,50).” We added the reference numbers 33,49,50 (lines 348-351).

- “In this investigation, the LDL-C levels in the intervention group decreased significantly after receiving the probiotics compared to the placebo one. High levels of LDL-C are a risk factor for diabetes mellitus and cardiovascular disease (49,58–60).” We added the references number 49,58-60 (lines 372-374).

- “The breakdown of bile salts in the digestive tract is attributed to BSH, which is another way in which probiotics can decrease cholesterol. It is then eliminated through the feces and is in an insoluble state. Consequently, the liver must extract more cholesterol to produce bile salts. This is an indirect method of decreasing cholesterol (65,66).” We added the reference numbers 65,66 (lines 384-388).

5. Explain what technique was used to randomize the study population. 

 Response: Details of the randomization technique have been added to the methods section for clarification, as follows.

- “Stratified permuted block randomization was utilized to assign the participants into two groups.” (lines 98-99)

6. Put references in lines 176 to 178, 517 to 522, and other paragraphs such as these that do not have bibliographic references.

 Response: Bibliographic references have been added to the mentioned lines and other relevant paragraphs to provide appropriate support for the statements made, as follows. 

- “The probiotic strains used in this test can reduce sugar and lipid levels in both in vitro and in animal models (27–29).” References 27-29 have been added to support the information presented. (lines 120-121)

- “The breakdown of bile salts in the digestive tract is attributed to BSH, which is another way in which probiotics can decrease cholesterol. It is then eliminated through the feces and is in an insoluble state. Consequently, the liver must extract more cholesterol to produce bile salts. This is an indirect method of decreasing cholesterol (65,66).” References 65 and 66 have been added to support the information presented. (line 388)

- “The gastrointestinal tract is colonized by the intestinal microbiota, a key environmental factor that directly influences host health and contributes to the exacerbation of various diseases (6–9). Dysbiosis, an imbalance in the microbiota, reduces its diversity and function, leading to metabolic issues (10). A high-fat diet can worsen dysbiosis by increasing circulating lipopolysaccharide (LPS) and lipid levels, as well as compromising the intestinal barrier (11).” References 6-9, 10, and 11 have been added to support the information presented. (lines 57, 58, and 59)

- “The key health benefits of probiotics include immune modulation, enhanced mineral and vitamin absorption, constipation relief, microbiome regulation, post-antibiotic microbiome stabilization, increased gastrointestinal resistance to pathogens, and pathogen reduction through short-chain fatty acid (SCFA) production (17–19).” References 18 and 19 have been added to support the information presented. (line 72)

- “Low-density lipoprotein cholesterol (LDL-C) is considered a critical contributor to the development of cardiovascular disease (33).” Reference 33 has been added to support the information presented. (line 167)

- “The QIAamp DNA Stool Mini Kit (Qiagen, USA) was used to extract DNA (35). The quantity and quality of the DNA were assessed using nanodrop and electrophoresis (36). By using 2X KAPA hot-start ready mix and 515 F and 806R primers, the V4 hypervariable region of the 16S rRNA gene was amplified by polymerase chain reaction (PCR) (37). References 35 -37 have been added to support the information presented.” (lines 182-185)

- “After eight cycles of the previously described PCR condition, the 16S amplicons were purified using AMPure XP beads and indexed with a Nextera XT Index Kit (38). Ultimately, the PCR products underwent cleaning and pooling in preparation for the Illumina® MiSeq™250-base paired-end read sequencing and cluster formation (39,40).” References 38 -40 have been added to support the information presented.” (lines 189-191)

- “The classify-sklearn naive Bayes taxonomy classifier was adopted to assign taxonomy to amplicon sequence variants (ASVs) based on comparison with the Greengenes 13_8 99% operational taxonomic units (OTUs) reference sequences (42). By applying the Kruskal-Wallis test and a permutational multivariate analysis of variance (PERMANOVA) with 999 permutations, respectively, statistical tests of the alpha and beta diversity were carried out (43). A heat tree analysis was generated for pairwise comparisons of the taxonomic differences between microbial communities with the MicrobiomeAnalyst web-based platform (44). ” References 42 -44 have been added to support the information presented.” (lines 202-206)

- “Stata/SE 16.1 software (StataCorp LP, College Station, TX, USA) was employed to analyze the statistical data (47).” Reference 47 has been added to support the information presented. (line 218)

- “The Benjamini-Hochberg (FDR) correction was applied to p-values for multiple hypothesis testing (48).” Reference 48 has been added to support the information presented. (line 225)

- “In this investigation, the LDL-C levels in the intervention group decreased significantly after receiving the probiotics compared to the placebo one. High levels of LDL-C are a risk factor for diabetes mellitus and cardiovascular disease (49,58–60).” References 49,58–60 have been added to support the information presented. (line 374)

- “In this research, the total cholesterol and triglyceride levels decreased significantly after the consumption of the probiotics for 12 weeks. Compared with pre-intervention placebo values, total cholesterol declined by 16.05% in the probiotic group versus 1.67% in placebo, while triglycerides fell by 7.04% versus 3.54%, respectively. Although HDL-C decreased in the probiotic group, the change was not statistically significant. These results align with previous trials: one involving dyslipidemic patients showed a 13.6% reduction in total cholesterol in the L. plantarum group after 12 weeks (67), and another by Keleszade et al. found that six weeks of L. plantarum ECGC 13110402 use significantly lowered total cholesterol and LDL-C levels (68). Additionally, B. longum CCFM1077 reduced total cholesterol by 13.87% in hyperlipidemic patients, and synbiotic foods containing L. sporogenes and inulin improved triglycerides and HDL-C (69). Overall, both fermented milk products and probiotics significantly reduced total cholesterol and LDL-C, with interventions exceeding four weeks proving more effective (70).” This paragraph has been revised for conciseness while maintaining the original content and references 67-70 have been added to support the information presented. (lines 389-400)

- “Probiotics could influence SBP through several mechanisms related to gut microbiome modulation (71). These outcomes are a result of the complex interactions that occur between the host's physiological processes, their metabolic products, and the gut microbiome (72). Certain probiotic strains can ferment and create peptides that mimic the effects of angiotensin-converting enzyme inhibitors, a type of medication used to treat hypertension (73).” References 71-73 have been added to support the information presented. (lines 402-405)

- “These clinical trials have investigated the impact of probiotics on blood pressure, with some reporting modest reductions in SBP reductions (reduced SBP by 3.05 mmHg and DBP by 1.51 mmHg), which is consistent with our work (reduced SBP by 10 mmHg with statistical significance, DBP 5 mmHg with no statistical significance) (75,76).” References 75,76 have been added to support the information presented. (line 417)

- “In this study, the LDL-C levels showed a marked decrease in the probiotic group compared to the placebo group, dropping by 23.37% and 2.94%, respectively, following the intervention. These results are supported by a meta-analysis of randomized controlled trials, which reported that consuming probiotic Lactobacillus strains, particularly L. reuteri and L. plantarum, significantly reduced both total cholesterol and LDL-C levels (69). In a controlled, randomized, double-blind trial conducted among dyslipidemic patients, supplementation with a combination of three L. plantarum strains (CECT 7527, CECT 7528, and CECT 7529) resulted in a 17.6% reduction in LDL-C levels after 12 weeks compared to the placebo group (67). In another randomized trial, the administration of Bifidobacterium longum CCFM1077 to hyperlipidemic patients reduced LDL-C levels by approximately 13.88%. This benefit may be attributed to the proliferation of anti-obesity-related bacterial genera and favorable shifts in fecal metabolite profiles (77).” References 69, 67, and 77 have been added to support the information presented. (lines 419-430)

- “Moreover, higher quantities of SCFAs can be produced more effectively by a more diversified gut microbiota (78,79).” References 78,79 have been added to support the information presented. (line 436)

- “The fermentation of complex carbohydrates produces metabolites known as SCFAs (80,81). Although butyrate can only be generated by members of the phylum Firmicutes, members of the Bacteroidetes phylum can also create acetate (82,83). Eubacterium rectale and Roseburia show a positive connection with SCFAs (84).” References 80-84 have been added to support the information presented. (lines 438-440)

- “SCFAs can be produced by lactic acid bacteria, such as those of the genera Lactobacillus and Bifidobacterium, even though these are not classified under the SCFAs category (88).” Reference 88 has been added to support the information presented. (line 450)

- “This imbalance is caused by an overgrowth of opportunistic bacteria and can lead to various health issues, including inflammatory bowel disease (IBD) (90), obesity (91), and metabolic disorders (92).” References 90-92 have been added to support the information presented. (lines 458-459)

- “Patients with autoimmune disorders (96), type 2 diabetes (97), and atherosclerotic diseases (98) frequently have abnormally low levels of SCFA-producing gut microbes (99).” References 96-99 have been added to support the information presented. (lines 470-471)

- “Therefore, creating and introducing bacterial occupants (or co-inhabitants) that have the ability to cross-feed and increase/reduce the number of SCFA-producing bacteria in the gut could be a viable probiotic strategy to aid in the treatment of several human illnesses (78,79).” References 78-79 have been added to support the information presented. (line 484)

- “Patients with obesity typically have altered gut microbiota compositions, suggesting that the gut microbiome may be a contributing component to the development of obesity (103,104).” References 103-104 have been added to support the information presented. (line 486)

- “An increased risk of cardiovascular disease may be closely linked to diabetes mellitus (109). A changed gut microbiome can lead to oxidative stress-related diseases, as well as obesity, metabolic endotoxemia, B-cell dysfunction, and systemic inflammation (110,111).” References 109-111 have been added to support the information presented. (lines 502-503)

- “One of the key components of lipid metabolism is RNA transport. The tissue-specific targeting of RNA-based therapies is now possible, increasing the safety of this treatment (115,116). RNA-based therapies have the potential to dramatically improve outcomes for individuals who have or are at risk for atherosclerotic cardiovascular disease, and multiple putative target proteins have been identified (117). The biosynthesis of unsaturated fatty acids, which are considered healthy fats, can lower blood cholesterol levels and decrease the risks of cardiovascular diseases (11

---

## [Decision Letter · Decision Letter 3]

23 Dec 2024

The efficacy of Lacticaseibacillus paracasei MSMC39-1 and Bifidobacterium animalis TA-1 probiotics in modulating gut microbiota and reducing the risk of the characteristics of metabolic syndrome: A randomized double-blinded, placebo-controlled study

PONE-D-24-31720R3

Dear Dr. Malai Taweechotipatr,

We’re pleased to inform you that your manuscript has been judged scientifically suitable for publication and will be formally accepted for publication once it meets all outstanding technical requirements.

Kind regards,

António Machado, PhD

Academic Editor

PLOS ONE

Additional Editor Comments (optional):

Reviewers' comments:

Reviewer's Responses to Questions

**Comments to the Author**

1. If the authors have adequately addressed your comments raised in a previous round of review and you feel that this manuscript is now acceptable for publication, you may indicate that here to bypass the “Comments to the Author” section, enter your conflict of interest statement in the “Confidential to Editor” section, and submit your "Accept" recommendation.

Reviewer #3: All comments have been addressed

2. Is the manuscript technically sound, and do the data support the conclusions?

Reviewer #3: Yes

3. Has the statistical analysis been performed appropriately and rigorously? 

Reviewer #3: Yes

4. Have the authors made all data underlying the findings in their manuscript fully available?

Reviewer #3: Yes

5. Is the manuscript presented in an intelligible fashion and written in standard English?

Reviewer #3: Yes

6. Review Comments to the Author

Reviewer #3: All comments have been addressed and authors should be complimented for performing such an original and relevant paper.

7. PLOS authors have the option to publish the peer review history of their article (what does this mean?). If published, this will include your full peer review and any attached files.

Reviewer #3: **Yes: **Fabrizio D'Ascenzo

---

## [Editor Report · Acceptance letter]

30 Dec 2024

PONE-D-24-31720R3 

PLOS ONE

Dear Dr. Taweechotipatr, 

I'm pleased to inform you that your manuscript has been deemed suitable for publication in PLOS ONE. Congratulations! Your manuscript is now being handed over to our production team.

Kind regards, 

on behalf of

Dr. António Machado 

Academic Editor

PLOS ONE